# Stochastic biological system-of-systems modelling for iPSC culture

Hua Zheng [1], Sarah W. Harcum [2✉], Jinxiang Pei[1] & Wei Xie [1✉]

Large-scale manufacturing of induced pluripotent stem cells (iPSCs) is essential for cell therapies and regenerative medicines. Yet, iPSCs form large cell aggregates in suspension bioreactors, resulting in insufficient nutrient supply and extra metabolic waste build-up for the cells located at the core. Since subtle changes in micro-environment can lead to a heterogeneous cell population, a novel Biological System-of-Systems (Bio-SoS) framework is proposed to model cell-to-cell interactions, spatial and metabolic heterogeneity, and cell response to micro-environmental variation. Building on stochastic metabolic reaction network, aggregation kinetics, and reaction-diffusion mechanisms, the Bio-SoS model characterizes causal interdependencies at individual cell, aggregate, and cell population levels. It has a modular design that enables data integration and improves predictions for different monolayer and aggregate culture processes. In addition, a variance decomposition analysis is derived to quantify the impact of factors (i.e., aggregate size) on cell product health and quality heterogeneity.

[1] Mechanical and Industrial Engineering, Northeastern University, Boston, MA 02115, USA. [2] Bioengineering, Clemson University, Clemson, SC, USA.
✉email: harcum@clemson.edu; w.xie@northeastern.edu

Since induced pluripotent stem cells (iPSCs) have the potential to differentiate into any cell type in the body, the discovery and availability of iPSCs have created numerous opportunities in the fields of regenerative medicines, cell therapies, drug discovery, and tissue engineering[1,2]. Large-scale manufacturing of iPSCs will be essential to support these clinical and research applications, not currently met to date. While iPSCs can be grown in small colonies in adherent monolayers (such as in petri-dish), this culture method is not ideal for large-scale manufacturing. Thus, stirred suspension cultures are recommended for manufacturing purposes[3,4]. Due to strong cell-to-cell interactions, one challenge that confronts large-scale iPSC cultures is the tendency of iPSCs to self-aggregate, which can lead to large aggregates in suspension bioreactors. Further, stirred suspension cultures can experience complex hydrodynamics conditions that can affect cell behaviors, including aggregation, metabolism, apoptosis, expansion, and differentiation[5].

As stem cells are highly sensitive to environmental conditions, a critical concern for iPSC aggregate cultures is spatial heterogeneity. Basically, nutrients and differentiation factors can become unevenly distributed in iPSC aggregates. These variable conditions across an aggregate can result in heterogeneous cell populations and cell death[6–8]. Therefore, understanding spatial heterogeneity and controlling aggregate size is crucial for predictable and consistent results in iPSC cultures.

Metabolic kinetic modeling is valuable to advance the scientific understanding of stem cell cultures, predict cell growth and functional behaviors, and guide feeding and agitation strategies. While several models have been developed to describe different aspects of iPSC cultures, a comprehensive mechanistic model that captures the multi-scale and heterogeneous nature of iPSC aggregate cultures is still lacking. For example, the Monod-type unstructured-unsegregated culture model developed by Galvanauskas et al.[9] assumes homogeneity of the entire iPSC population and does not account for intracellular variability. The population balance model (PBM) proposed by Bartolini et al.[10] focuses on the temporal evolution of the size distribution of embryonic stem cell (ESC) aggregates, while Van Winkle et al.[6] modeled a single human ESC (hESC) spherical aggregate, yet neglected the dynamics of the cell population. Recently, Odenwelder et al.[11] applied a metabolic flux analysis (MFA)[12,13] to describe the effects of glucose and lactate concentrations on monolayer iPSCs. Wu et al.[14] developed a mechanistic model that described cell aggregation and considered oxygen transport for iPSC aggregate cultures, however, it neglected metabolite diffusion, intracellular metabolism, and cell metabolic heterogeneity.

Existing mechanistic models for mammalian cell cultures often assume the cell population is homogenous. Thus, these metabolic models overlook the stochastic nature of living cells. For cultures such as Chinese hamster ovary (CHO) cells and yeast, the assumption of spatial homogeneity may be sufficient in well-stirred systems where cell aggregates are non-existent or only involve a few cells. In these cases, unsegregated models, such as dynamic flux balance analyses[12,15] and cell culture kinetic models[16] have been demonstrated to predict culture performance. But unlike single-cell suspension cultures, aggregate cultures, common to iPSCs, are not homogenous. As the aggregate size increases, spatial heterogeneity likely increases cell-to-cell variation and increases metabolic heterogeneity. While stochastic chemical kinetics[17,18] and queueing network models[19,20] have been developed to describe the dynamics and inherent stochasticity of chemically reacting systems and metabolic networks, these approaches do not account for the effect of cell aggregation on metabolite variations and potential cell heterogeneity common to iPSC cultures. Additionally, spatial variance component analysis (SVCA) was introduced to study spatial heterogeneity

contributed by cell-to-cell interactions, intrinsic effect, and environmental effect[21]. However, this study is built on the random effect model and it is challenging to faithfully characterize the underlying complex interaction mechanisms and causal interdependencies from molecular to cellular to macroscopic levels.

In a broader scope, several multiscale bioprocess models have been developed for studying aggregate structures in clinical studies, such as cancer tissues[22,23], and biofilms[24]. Notable examples include software packages like PhysiCell[25], Chaste[26], ChemChaste[27], BMX[28], and Morpheus[29]. These tools fall under the category of agent-based models, which integrate cell-based metabolic reactions, reaction-diffusion processes, and individual cell growth dynamics. Furthermore, these models have the potential to include specialized stochastic cell models, allowing them to represent regulatory structures and account for mechanical and chemical interactions. While these methods could be effective for modeling small aggregate systems, these models become computationally prohibitive when applied to large-scale iPSC manufacturing processes.

To overcome the limitations of existing methods, this paper presents a novel biological systems-of-system (Bio-SoS) model with modular design to characterize the mechanisms in iPSC aggregate cultures and describe the spatial-temporal causal interdependencies from individual cells to cell aggregates and to cell populations. It is built on mechanistic modules, including (1) a stochastic metabolic network (SMN) model describing cell metabolic response to environmental variation; (2) a PBM describing the iPSC proliferation, collision, and aggregation process due to cell-to-cell interactions; and (3) a reaction–diffusion model (RDM) characterizing spatial heterogeneity of micro-environmental conditions which accounts for the different diffusion rates of nutrient and metabolite molecules through cell aggregates. The modular design allows us to organically assemble individual mechanistic modules to construct a Bio-SoS model for iPSC aggregate cultures, which facilitates the integration of data and information collected from different cell culture processes with different dynamics and spatial heterogeneous micro-environmental conditions. This Bio-SoS modeling philosophy for multi-scale bioprocess is extendable and applicable to general biological ecosystems, accounting for complex interactions and inherent stochasticity.

The proposed Bio-SoS framework represents a novel in-silico process analytical technology (PAT) for iPSC aggregate cultures, offering valuable insights into individual cell responses to micro-environmental changes and metabolic information across distinct aggregate locations. The key contributions are threefold. First, the proposed multi-scale Bio-SoS model has a modular design that facilitates the integration of data from different culture conditions (such as 2D monolayer cultures used in the lab and aggregate cultures recommended for industrial manufacturing). This modular design will accelerate iPSC's large-scale manufacturing process development without conducting extensive experiments. It can provide a valuable tool for yield optimization and cell product quality consistency control. Second, since the objective for iPSC cultures is undifferentiated biomass, i.e., the cells are the product, the proposed variance decomposition analysis on the Bio-SoS mechanistic model enhances our systematic understanding of iPSC culture spatial heterogeneity, predicts the impact of critical factors (i.e., aggregate size) on metabolic heterogeneity, and enables us to avoid unwanted cell death or heterogeneous cell populations during expansion. It can identify the root causes of cell-to-cell variation, analyze intracellular metabolism at different positions within an aggregate, and determine the optimal aggregate size range across different bioreactor conditions. Third, the proposed Bio-SoS simulation provides a comprehensive and

efficient way to account for the complex and stochastic nature of iPSC aggregate and monolayer cultures. In contrast to existing agent-based simulation tools[22–29], the Bio-SoS model is a specialized simulation tool tailored for iPSC cultures. It can improve simulation efficiency through (1) modeling iPSC aggregation using population balance equations to ensure a more efficient representation of cellular dynamics during aggregation; (2) constructing a coarse-grained approach that divides each aggregate into small spherical shells and assumes cellular metabolisms and micro-environment are homogeneous in each spherical shell; and (3) utilizing a single-cell SMN model to predict the cell metabolic response to environmental variation.

## Results

**Bio-SoS model for multi-scale iPSC cultures.** Within (3-dimensional or 3D) aggregate cultures, cells proliferate and interact with each other at three levels: intracellular metabolic reactions, metabolites and nutrients diffusion through the cell aggregates, and aggregate interactions within bulk culture fluid. Each individual cell within the culture is a complex system with potentially stochastic behavior. As these cells cluster together, a larger system of systems is formed with micro-environmental conditions shaped by cell interactions. Taken together, aggregates comprise the entire iPSC population in the bioreactor interacting with the bulk media.

The proposed Bio-SoS model characterizes the complex interactions and regulatory reaction network mechanisms from molecular- to cellular-, and to macro-kinetics; see Fig. 1. Based on the causal interdependencies between these biological systems, the Bio-SoS model was constructed incorporating three interconnected mechanistic modules summarized below (see "Methods" section for details). The intracellular regulatory metabolic reaction network is connected to the aggregate via the transport of metabolites across the cellular membrane, while cells within an aggregate are linked through the diffusion of intra-aggregate metabolites and nutrient supplies. This Bio-SoS reaction network mechanism for iPSC aggregate culture is characterized through the integration of an RDM and single-cell SMNs, quantifying the metabolic heterogeneity. The aggregation process of the cell population is effectively characterized using PBM. The Bio-SoS model can both sample and computationally efficiently characterize the spatial heterogeneity in micro-environmental conditions and the variability in cell-to-cell metabolism.

1. *PBM*. A PBM is used to describe the aggregation process accounting for cell proliferation and coalescence/collisions of iPSC aggregates.
2. *RDM*. It is constructed to describe cell-to-cell interactions and the intra-aggregate fluid dynamics, which is a fundamental process involving the transport of reacting nutrients and metabolites through iPSC aggregates. This movement is influenced by the metabolite production/consumption and a diffusive flux that is proportional to the local concentration gradient. By considering the complex interplay between individual cell metabolic reactions and their micro-environment, this model can estimate how long cells might experience a particular condition. The diffusion coefficients used in this study are listed in Supplementary Table 4.

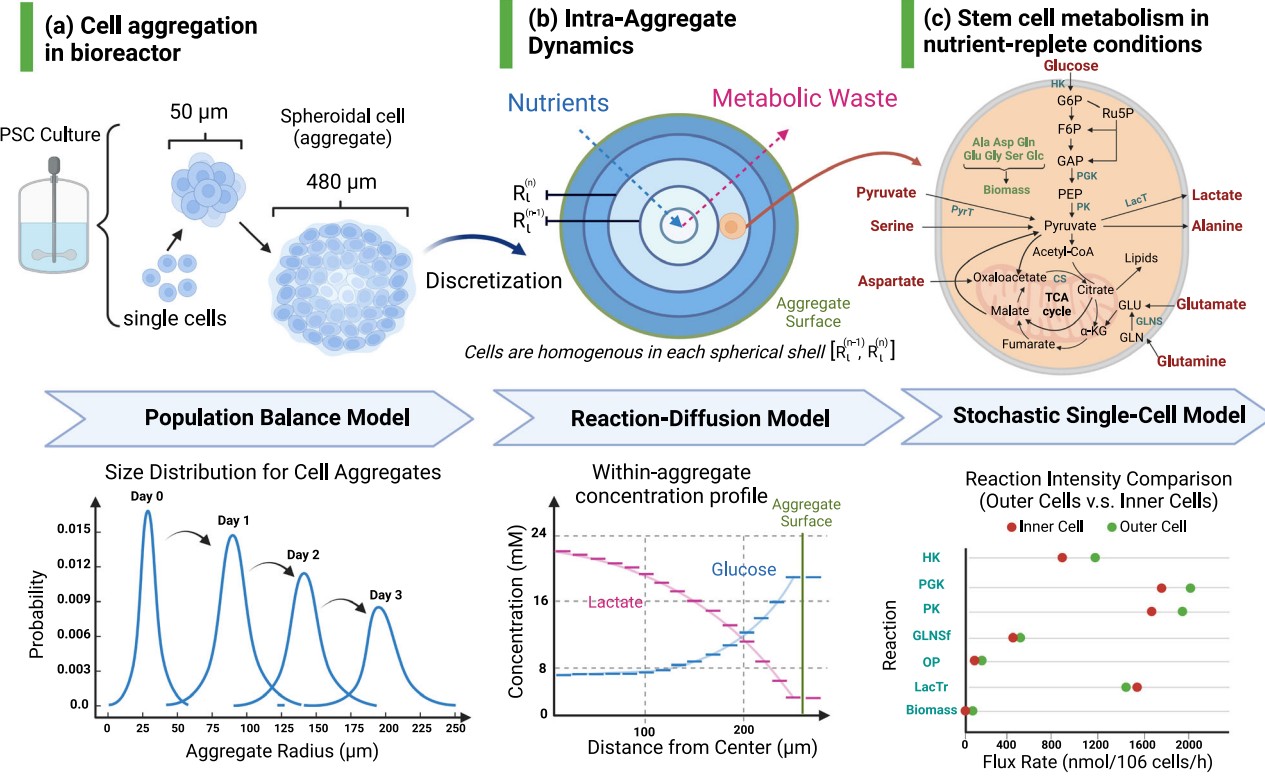

**Fig. 1 An illustration of the proposed Bio-SoS for iPSC aggregate culture (Created with BioRender.com). a** The sizes of cell aggregates grow over culture time. The PBM was used to describe the dynamics of the cell aggregation process and predict the aggregate size distribution. **b** Cell aggregates are spatially decomposed into equally sized small spherical shells. In each shell, the metabolic concentrations are assumed to be homogenous. RDM was used to characterize the intra-aggregate diffusion dynamics of nutrient and metabolite concentrations, showing that the nutrient concentrations increased as the distance from the center increased, while the metabolic waste concentrations decreased from the center. **c** The SMN is built to describe the intracellular reactions to microenvironmental perturbations. Suppose cells residing in each spherical shell are homogeneous. The metabolic heterogeneity is characterized by the difference in fluxes between exterior cells and inner cells located at different positions in aggregates.

3. *SMN for single cells*. The metabolic molecular reaction network is constructed from the curated biochemical interactions based on experimental data[11]. These interactions characterize intracellular metabolisms and cellular responses to environmental perturbations. By following the studies on stochastic molecular reaction models[19,20], a Poisson process is utilized to model molecular enzymatic reaction occurrence within the SMN to capture the random nature of enzyme-substrate collisions and subsequent reactions. The expected metabolic reaction rates for homogeneous cell population were derived from Wang et al.[30] and calibrated using 2D monolayer culture experimental data of extracellular metabolites and intracellular isotopic measurements from Odenwelder et al.[11]. The iPSC SMN (composed of 32 metabolites and 38 reactions) mainly focuses on central carbon metabolism and contains the major reactions for glucose consumption, the TCA cycle, anaplerosis, pentose phosphate pathway (PPP), and amino acid metabolism. The reactions, metabolites, and enzymes that are used in this study are listed in Supplementary Tables 1–3.

Built on the Bio-SoS model characterizing the causal interdependencies of iPSC aggregate culture, a spatial variance analysis approach is derived to study the root cause of cell metabolic heterogeneity (see "Methods" section for details), which can be used to guide the control of iPSC cultures, including aggregate size. Because the system comprises aggregates of multiple sizes, cell aggregates were numerically classified into $L$ groups. To further consider the spatial heterogeneity in each aggregate with radius, say $R_\ell$ for $\ell = 1, 2, \ldots, L$, the aggregates were divided into spherical shells; see the illustration of cells located in the $n$th spherical shell $[R_\ell^{n-1}, R_\ell^n]$ in Fig. 1b. The proposed Bio-SoS variance analysis approach can assess the contribution of spatial heterogeneity from aggregates with different sizes to the cell product metabolic variation and quantify the metabolic heterogeneity through studying the relative changes of metabolic fluxes for cells residing at different locations and time within an aggregate.

**Model validation for Bio-SoS framework**. The three individual model modules (i.e., PBM, RDM, and SMN) were validated with experimental data from literature[7,11,14]; then the integrated Bio-SoS model was validated based on both monolayer culture data from Odenwelder et al.[11] and aggregate culture data from Kwok et al.[7].

First, the PBM was validated using cell proliferation and aggregation dynamics data from Wu et al.[14]. The experimental and model-predicted aggregate growth profiles are shown in Fig. 2a and the time-varying aggregate size distribution is illustrated in Fig. 2b. Overall, the PBM captured the aggregation dynamics. The model-based prediction is obtained by solving the PBM numerically and by using finite differences over equally spaced intervals (i.e., 1 μm) in the radius domain of an aggregate and 0.1 h in the time domain[31].

Second, the RDM was validated by using stirred-tank suspension bioreactor data from Wu et al.[14]. The transport of metabolites and nutrients through 3D aggregates is crucial for the effectiveness of cultivation systems. This is especially significant in stem cell cultures, as cell metabolism determines iPSC product functional quality attributes, and it plays an important role in determining pluripotency and lineage specification[32]. Given the stem cell aggregate property (i.e., porosity and tortuosity) estimated based on the measures in Wu et al.[14], the RDM was solved analytically by applying a local quasi-steady-state approximation, i.e., aggregates are spatially decomposed into equally

sized small spherical shells and the metabolic concentrations are assumed to be homogenous within each shell. Then, the model predicted profiles of heterogeneous metabolite concentrations under steady-state conditions, which are shown in Fig. 2c and Supplementary Fig. 2. It should be noted that these results were obtained from the single representative aggregate simulation. Clearly, the diffusion of nutrients (i.e., glucose, glutamine, and serine) becomes limited as the aggregate radius increases, leading to reduced nutrient levels for cells residing closer to the core. For large aggregates, the cells in the inner area are starving due to a shortage of glucose, glutamine, and serine. This finding is consistent with the glucose transport limitation in human mesenchymal stem cells reported by Zhong et al.[33]. Also, limited diffusion resulted in higher concentration of metabolite wastes and inhibitors (i.e., lactate, ammonium, and glutamate) in the aggregates as the size increases. Future investigations of nutrient and metabolic waste levels within aggregates using imaging techniques could provide more evidence.

Third, the SRN was used to characterize single-cell metabolism with the backbone mean metabolic reaction rate for a homogeneous cell population characterized by the deterministic mechanistic model[30]. To support the prediction of iPSC response to micro-environmental perturbations occurring during the aggregate culture process, the SRN model was estimated and validated by using the time-course data of K3 iPSCs cultured in monolayer under high/low nutrient and metabolic waste concentrations as described in Odenwelder et al.[11]. Since induced pluripotent stem (iPS) cells cultured in petri-dish 2D monolayer are homogeneous, the deterministic mechanistic model was derived to characterize the expected metabolic flux rate response for cell population in Wang et al.[30]. Then, to simulate a monolayer culture, the radii of cell aggregates were set to a uniform 7.5 μm, which corresponds to the single cell radius[14]. In Fig. 3, the bulk metabolite concentrations predicted from the Bio-SoS for homogeneous iPS cell population align with both the measured values under a representative high glucose and low lactate experimental condition. As we expected, the simulation results of Bio-SoS closely resemble those of the traditional MFA and deterministic mechanistic models when no aggregation and homogeneous cell population are considered.

In addition to being able to model the monolayer culture behavior, we sought to further validate the Bio-SoS model on aggregate cultures. As mentioned earlier, the Bio-SoS model was developed and trained by utilizing monolayer culture data from K3 iPSCs[11] and cell aggregation profiles from hESCs[14]. To test its extrapolation prediction performance, we used a collection of aggregate culture data of FSiPS (short for FS hiPSC clone 2) collected from stirred suspension bioreactor from Kwok et al.[7] and followed the same culture protocol implemented in Kwok et al.[7]. To visually compare the measured and predicted glucose consumption and lactate production, we rescaled the vertical axes while keeping the values unchanged and presented the results in Fig. 4. The Bio-SoS model demonstrates good prediction performance on both glucose and lactate concentrations (represented by the red line) as the predictions are within the 95% confidence intervals of the measured values (represented by the blue error bars). Therefore, the model's prediction using the bioreactor data from FSiPS provides additional confidence in the reliability of the proposed Bio-SoS model. This also suggests its potential for use with other cell lines.

Despite the differences between the cell line and culture conditions between the monolayer training dataset and aggregate cultures, the Bio-SoS model provided meaningful insights into metabolic behaviors. The reliability of extrapolated predictions using the Bio-SoS model is based on key insights included in the model design. First, the underlying metabolic pathways and

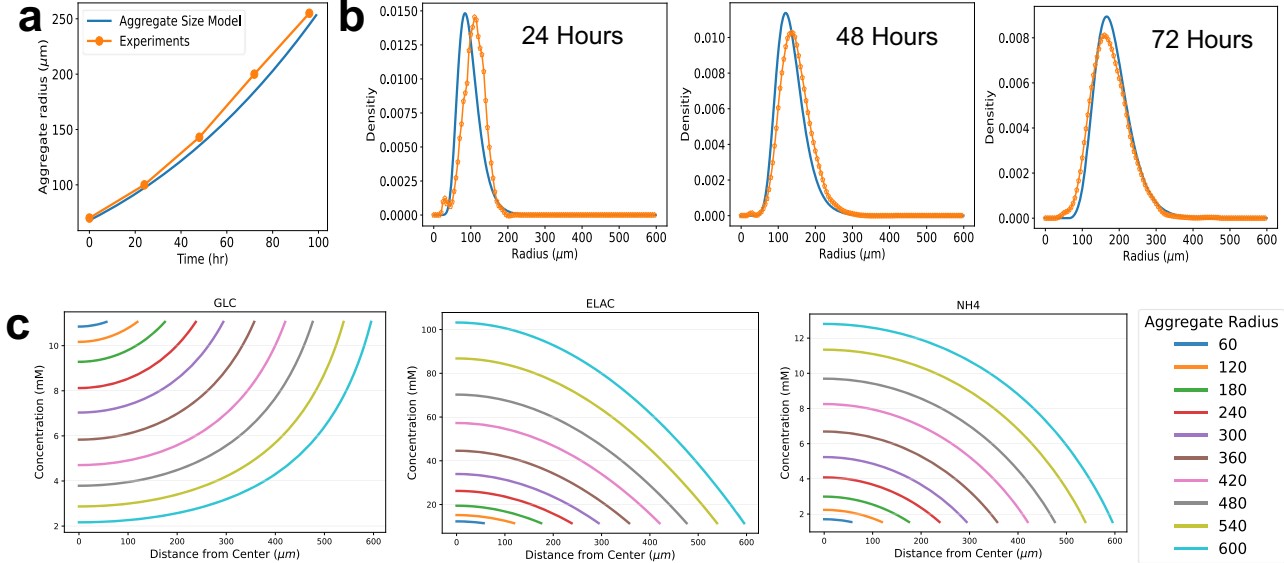

**Fig. 2 PBM and RDM models to predict iPSC aggregation process in stirred suspension bioreactor cultures.** Panel **a** shows the experimental and the predicted aggregate size; Panels **b** shows representative experimental data and PBM predictions of size (radius) distribution for cell aggregates, denoted by $\phi_R(R, t)$, at 24, 48 and 72 h (see "Methods" section for details). The dotted orange lines represent the experimental data from Wu et al.[14], while the blue lines show the PBM size distributions. **c** Steady-state concentration profiles of Glucose, Lactate, and Alanine for iPSC aggregates with radii ranging from 60 to 600 μm, each with identical values for diffusion coefficients $D_i^a$, porosity $\varepsilon = 0.27$ and tortuosity $\tau = 1.5$ and initial intracellular metabolite concentrations.

cellular processes remain relatively consistent across different cell lines and cell types. These simulation results provide clear evidence that the fundamental principles governing cellular metabolism and nutrient utilization persist, even when cells are cultivated under different conditions. By incorporating this fundamental metabolic information into the model, it can capture the core metabolic behaviors and predict metabolic outcomes across different micro-environmental conditions. Second, the model considers the diffusion of metabolites and accounts for the spatial distribution of nutrients within the culture system. This incorporation allows for a more realistic representation of nutrient availability and metabolic interactions within the aggregates.

**In-silico study of iPSC aggregate health conditions.** Aggregate size influences the transport of nutrients, oxygen, metabolites, and growth factors due to the diffusion of each species, which is related to molecular size and charge. Cells inside large aggregates potentially experience hypoxic conditions and poor nutrient supply due to the limited diffusion of oxygen and nutrients from the bulk media to the center. As a result, the cell growth is reduced. It has already been reported that oxygen concentration in the center region of larger embryoid bodies (400 μm in diameter) is 50% lower compared to in medium embryoid bodies (200 μm in diameter), which caused apoptosis at the core due to low oxygen diffusion[6,33,34]. Further, low diffusion can limit the removal of waste metabolites, such as lactate and ammonia, and increase cell necrosis in the center region as these species reach critical levels.

Previous studies have shown that high lactate concentration can have negative effects on stem cell pluripotency. For example, murine ESC (mESCs) and iPSCs proliferated and maintained pluripotency in lactate concentrations up to 40 mM[35], while hESCs exhibited decreased pluripotency through Tra-1-60 expression after continuous passaging in 22 mM lactate-containing media[36]. Ouyang et al.[37] showed that mESCs are extremely sensitive to the presence of lactate in media. They

inferred that the growth of mESC was inhibited at lactate greater than 16 mM and that high lactate affected the cell pluripotency. Glucose has also been observed to affect the embryoid body formation potential of mESC when the concentration is less than 2.5 mM[35]. Therefore, in this study, we define cells as being unhealthy if glucose is less than 2.5 mM or lactate is greater than 40 mM.

The Bio-SoS model was used to predict the fraction of unhealthy cells using glucose and lactate concentration criteria (glucose < 2.5 mM or lactate > 40 mM) for aggregates of radius ranging from 30 to 600 μm. Figure 5a shows the percentages of unhealthy cells within an aggregate with a particular radius with varying bulk glucose concentration and a fixed lactate concentration of 0 mM. Notably, all aggregates, regardless of size, exhibited 100% of cells being classified as unhealthy when the bulk glucose concentration fell below 2.5 mM. In contrast, aggregates of radii less than 150 μm had higher fractions of healthy cells for bulk glucose concentrations greater than 5 mM. Figure 5b shows the percentage of unhealthy cells within an aggregate with a particular radius with varying bulk lactate concentration and a fixed glucose concentration of 20 mM. A different pattern was observed for cell health when the bulk lactate concentration changed. The fraction of unhealthy cells increases with the increased aggregate size, and it appears that cells are more sensitive to higher lactate concentrations compared to lower glucose concentrations.

**In-silico study of cell-to-cell metabolic heterogeneity.** It is well accepted that culture conditions need to remain uniform for optimal iPSC metabolic function and pluripotency maintenance. The formation of large aggregates increases the risk of heterogeneity due to the limited diffusion of nutrients and growth factors and the removal of waste metabolites. In order to describe the expected metabolic flux rate response of a homogeneous cell population to environmental change, the metabolic regulatory networks were adapted from a companion work by Wang et al.[30]. Supplementary Fig. 1 shows this metabolic regulatory network,

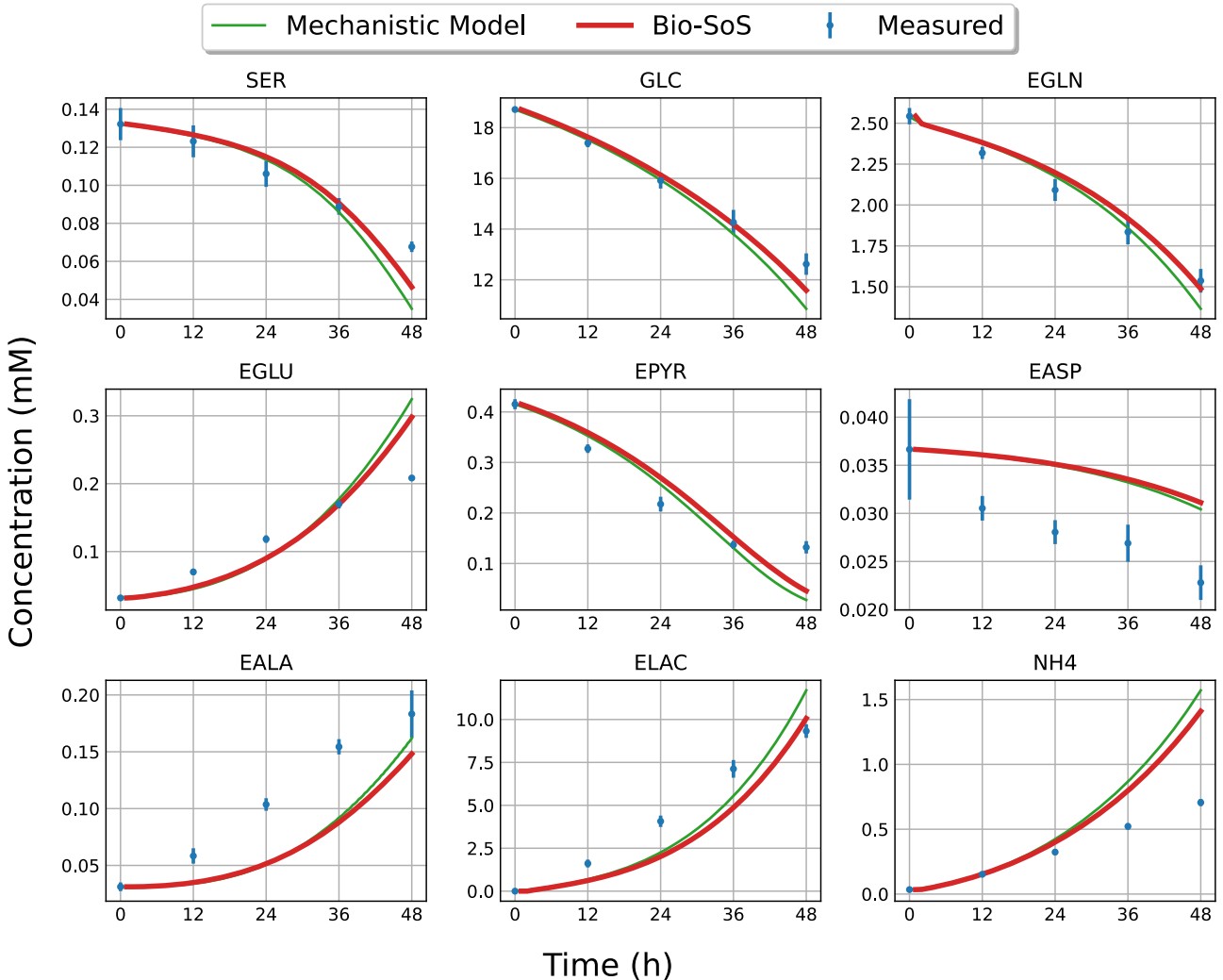

**Fig. 3 Comparison of the prediction obtained by using both deterministic mechanistic model[30] and the Bio-SoS model on the mean response for monolayer cultures of homogeneous K3 iPS cell population.** Monolayer experimental data are shown for bulk metabolites (mean ± SD). The green lines represent the predicted metabolite concentrations of the mechanistic model from Wang et al.[30]; the red lines represent the predicted metabolite concentrations of the Bio-SoS model. Blue circles with error bars represent experimental data. The description of metabolite abbreviations is in Supplementary Table 2.

which includes the major metabolic pathways such as glycolysis, TCA cycle, anaplerosis, PPP, and amino acid metabolism. The reactions of PPP were simplified to two reactions (i.e., Nos. 9 and 10 reactions in Supplementary Table 1) representing the oxidative phase/branch and non-oxidative phase/branch, respectively. The descriptions of metabolites and enzymes are organized according to the Enzyme Commission Numbers (EC-No.) and are listed in Supplementary Tables 2 and 3, respectively. By following the recent studies on stochastic molecular reaction network[19,20], we construct an SMN for single cells that can characterize the stochastic reaction network for individual cells and cell metabolic response to environmental change.

To understand how cell metabolism is affected by the aggregate size and micro-environmental changes, the metabolic reaction intracellular flux rates that are sensitive to the aggregate size are shown in Fig. 6a. Further, the key extracellular metabolite concentrations are shown in Fig. 6b. The flux rates and metabolite concentrations were standardized by subtracting the mean and dividing by the standard deviation calculated over all aggregates (see "Method" section). The expected flux rates of the (forward) reactions (e.g., GLNSf and HK) decreased gradually as the aggregate sizes increased, and the flux rate of the reverse reaction

(e.g., GLNSr) increased as the aggregate sizes increased. The biomass flux was observed to be relatively high for small aggregates, suggesting such aggregates provide favorable metabolic conditions for biomass production. The reversible reactions such as GLNS, LacT, GLDH, and ASTA were affected by the extracellular metabolite concentrations. For example, the flux of LacTr increases in large aggregates due to the high extracellular lactate levels while the fluxes of GLNSr and GLDHr increase due to low-glutamine levels inside large aggregates (Fig. 6b). In summary, these results confirm the importance of maintaining aggregate size for optimal biomass production. It has been observed previously that aggregates exceeding a diameter of 300 μm experience hypoxia and low core nutrient concentrations, resulting in cell necrosis and loss of pluripotency[38].

We further investigated the metabolic heterogeneity between inner and outer cells within aggregates of radii ranging from 60 to 360 μm (Fig. 7). The blue dashed line depicts the flux rate at the outer cell, while the colored dots represent the relative flux rate of inner cells at various locations. This ratio was calculated as the flux rate of the inner cell relative to the outer cell. Figure 7 shows the metabolic heterogeneity of the inner cells compared to outer cells for aggregates of various sizes. The greater deviation of larger

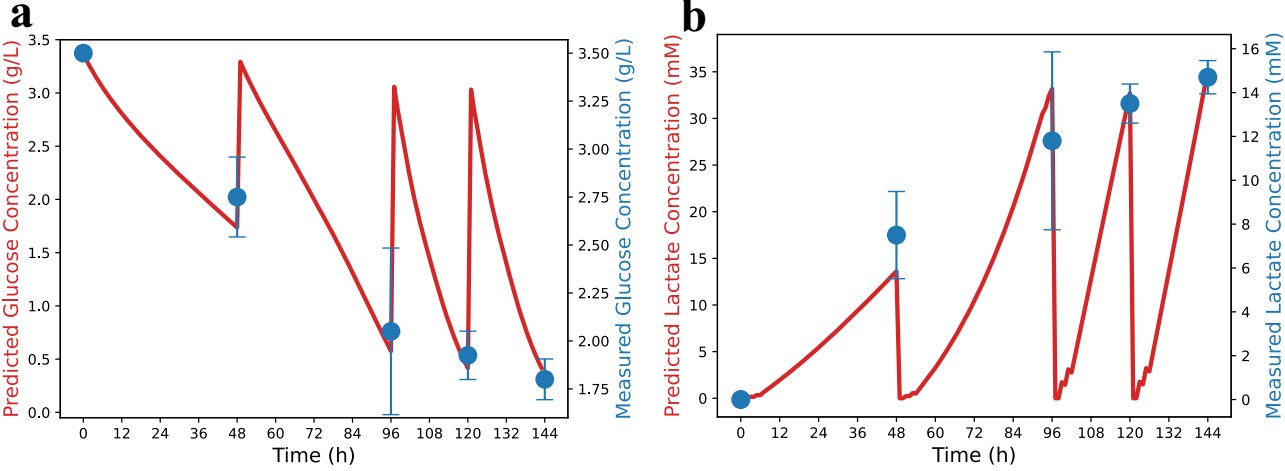

**Fig. 4 Glucose and lactate concentration profiles predicted by the Bio-SoS model in comparison with the literature data collected from stirred-tank suspension bioreactor. a** Glucose **b** Lactate. The Bio-SoS model (red line) was trained only with the monolayer data for K3 iPSC described in Odenwelder et al.[11]. The experimental data shown (blue circle, mean ± SD) are for FSiPS cultured in aggregate and described in Kwok et al.[7]. Note the trends predicted by the Bio-SoS model, which was trained on monolayer data, match the observed trends of aggregate cultures.

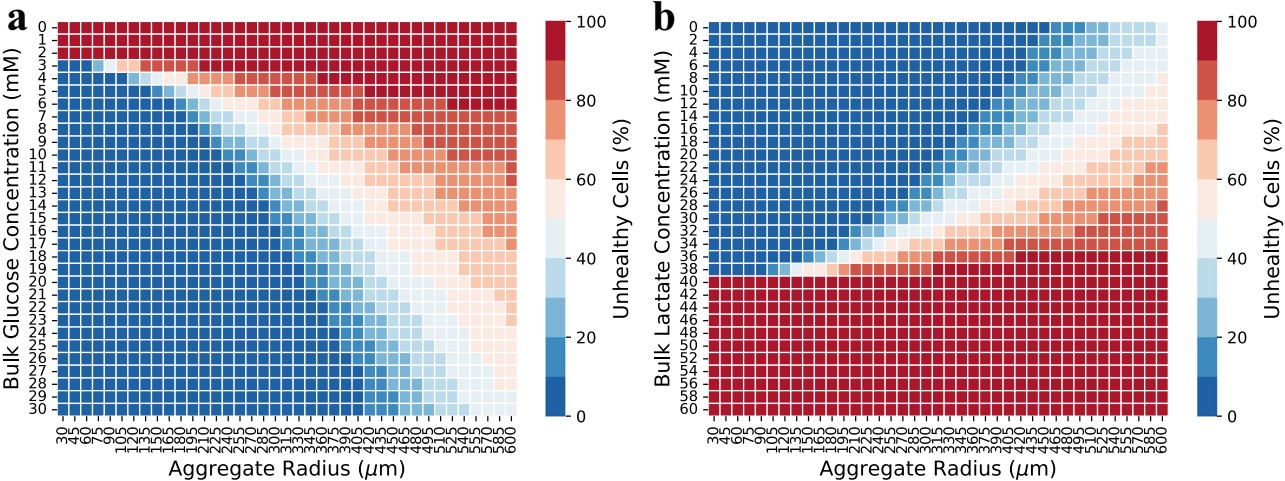

**Fig. 5 Unhealthy cells as a function of aggregate radius and metabolite concentration. a** Effect of the glucose concentration and aggregate radius on the percentage of unhealthy cells at a fixed bulk lactate concentration of 0 mM. **b** Effect of the lactate concentration and aggregate radius on the percentage of unhealthy cells at a fixed bulk glucose concentration of 20 mM. Results are averaged by 30 simulation runs.

aggregates (represented by purple dots) highlights that they experience greater heterogeneity, while the smaller aggregates (represented by orange dots) have more consistent metabolism.

The results in Fig. 7 also show a significant increase in metabolic heterogeneity from 24 to 48 h, as greater flux deviations were observed in the latter period. This can be attributed to the relative shortage of intracellular metabolites after 48 h. For example, the 48-h serine consumption flux (SAL) becomes more diverse due to low serine, while the poor nutrient supply affected biomass production leading to heterogeneous biomass fluxes (see Supplementary Fig. 1). To quantitatively assess the overall metabolic heterogeneity of each aggregate, the difference between flux rates of the inner and outer cells was calculated (Supplementary Tables 5 and 6). Based on the simulation results, the metabolic heterogeneity of a 240-μm aggregate was found to be approximately 14 times greater than that of a 60-μm aggregate. After 48 h, both aggregate sizes had doubled in metabolic heterogeneity. These simulations demonstrate that metabolic heterogeneity increases with both aggregate size and culture duration.

It is also worth noting that the TCA (tricarboxylic acid) cycle has been observed to maintain a stable flux in cell aggregates of different sizes, as shown in Fig. 7 (see the reactions highlighted in blue). This observation supports the widely accepted understanding that the TCA cycle is a fundamental housekeeping metabolic pathway (i.e., the flux rate maintains relatively stable in different conditions)[39] and that it is tightly regulated[40].

**Optimal aggregate size**. An important factor that needs to be strictly controlled in bioreactors is the aggregate size. If iPSC aggregates become too large, uneven diffusion of nutrients and growth factors can occur, causing cell death or heterogeneous cell populations. The optimal aggregate size can be determined by considering the balance between metabolic heterogeneity and biomass yield.

Simulations of a batch bioreactor were performed to calculate the mean and relative standard deviation (RSD) for biomass productivity (Fig. 8a–c) for a 72-h culture. Supplementary Fig. 3 presents additional simulation results for the mean and RSD of biomass productivity at 6-h intervals. These results indicated a

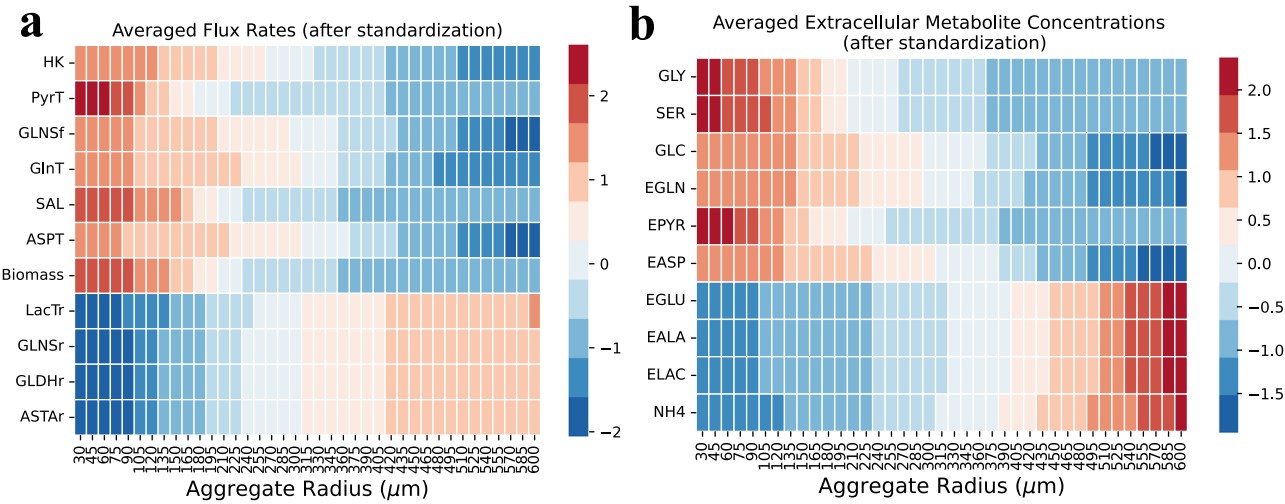

**Fig. 6 Expected reaction flux rates and extracellular metabolite concentrations by aggregate size. a** Standardized flux rate. **b** Standardized metabolite concentrations. Simulations were conducted with constant bulk glucose (25 mM), lactate (5 mM), and alanine (0.1 mM) concentrations. To facilitate visual comparison, all flux rates and concentrations were standardized by subtracting the mean and dividing the standard deviation calculated over all aggregates. Aggregates range from 30 to 600 μm. Results are averaged by 30 simulation runs.

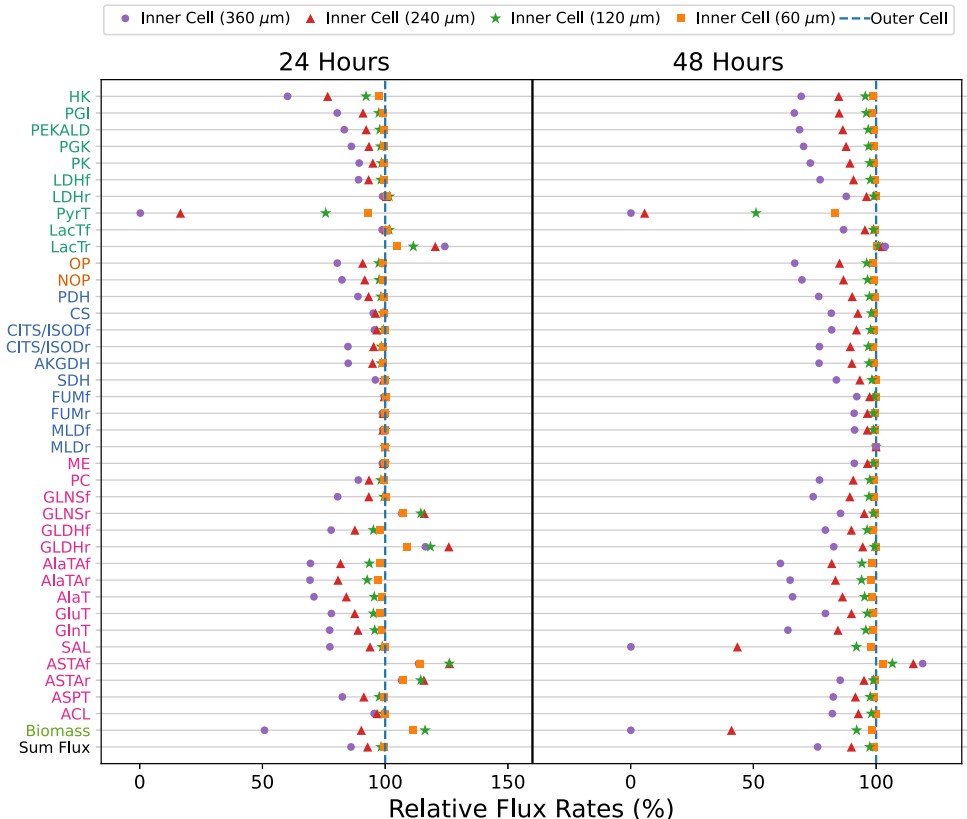

**Fig. 7 Metabolic heterogeneity of aggregates of varying sizes (60, 120, 240, and 360 μm in radius) at 24 and 48 h.** Specifically, the relative fluxes were compared for cells located at the center of the aggregates of various sizes, called the inner cells. The blue dashed lines represent relative fluxes of outer cells, which were consistent across all four aggregate sizes. The circle (purple) represents the relative flux of the inner cells for the 360 μm aggregates; the triangle (red) represents the 240 μm aggregate; the star (green) represents the 120 μm aggregate; the square (orange) represents that for 60 μm aggregate; normalized to the fluxes of outer cells. Results are averaged by 30 simulation runs.

yield-heterogeneity trade-off, as the mean biomass productivity consistently decreased while the RSDs consistently increased with increased aggregate size. It was also observed that the RSDs consistently increased with the culture time. During the initial 24 h, the majority of aggregates in the bioreactors exhibited

relatively stable biomass productivity (Fig. 8a). The aggregate size distribution (Fig. 2b) indicated that the number of aggregates with a radius greater than 200 μm was very low. This observation implies that the heterogeneity within the cell population is limited early in cultures. However, after 48 h, the culture entered a steady

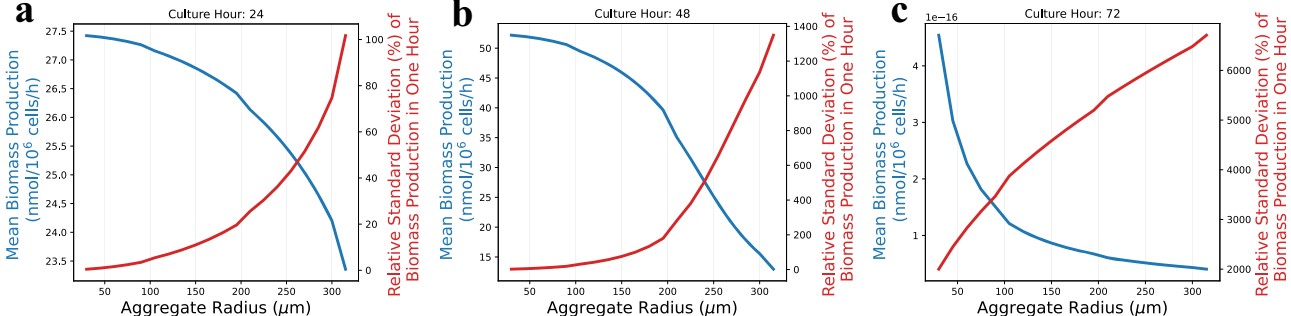

**Fig. 8 Biomass yield and variance decomposition analysis.** In panel, we simulated the biomass production of different aggregates (ranging from 30 to 300 μm in radius) for a duration of one hour with 100 replicates. The recorded results in panels **a**–**c** correspond to 24-, 48-, and 72-h cultures, respectively. The blue line represents the mean, and the red line represents the RSD of the biomass, i.e., $RSD_{biomass,\ell,t}$ with $\ell=30, 45, \ldots, 300$ (see Eq. (11) in "Methods" section for details).

state (as reported previously[11,30]). Figure 8b shows increased heterogeneity of biomass yield for larger aggregates. Notably, there was a distinct cut-off radius of 150 μm after 48-h culture. For a larger radius, the RSD increased significantly faster. These findings suggest the presence of an optimal aggregate size range that minimizes biomass productivity variability and aggregate heterogeneity. Lastly, as Fig. 8c suggests, the biomass productivity decreased significantly as the available nutrients were nearly depleted around 72 h. Overall, aggregates with a radius below 150 μm had relatively high biomass productivity and a relatively low heterogeneity for 48-h cultures. These simulation results suggest that iPSC cultures should be maintained as uniform aggregates around 150 μm in radius and should not be greater than 200 μm in radius. This evidence agrees with experiment observations in the literature[6,8,38].

## Discussion
The introduction of the Bio-SoS framework marks an important step in the development of multi-scale bioprocess mechanistic models and analytical technology for iPSC cultures. By effectively characterizing cell-to-cell interactions and complex mechanisms of iPSC aggregate culture, the proposed Bio-SoS not only supports data integration and enables the prediction of metabolic dynamics at different scales but also models spatial heterogeneity and quantifies metabolic intensities and variations across different aggregate locations and time. The model's versatility is shown by its ability to study cell health in different-sized aggregates, predict micro-environmental profile metabolite concentrations under diverse conditions, and determine the optimal aggregate size range and feeding strategy for maximum bioreactor efficiency. Ultimately, the proposed Bio-SoS presents a promising pathway towards low-cost and high-quality personalized cell therapies and offers a platform for optimizing large-scale iPSC manufacturing without extensive experiments.

The model validation study demonstrated that the proposed Bio-SoS model has good extrapolation prediction performance. Even though only the monolayer culture data of K3 iPSC with various initial conditions was used, the model was able to predict the metabolic dynamics for a different cell line (FSiPS) cultured in aggregates. This demonstrates the potential for transferring the learning from a monolayer culture to a stirred-suspension aggregate culture. From the methodological perspective, this success relies on the proposed biological system-of-systems modeling principle that can organically assemble a single-cell model to construct a Bio-SoS model for iPSC aggregate cultures, characterizing cell-to-cell interactions and cell response to spatial heterogeneous micro-environment conditions.

The versatility of the proposed framework is demonstrated by three in-silico scenarios. First, the Bio-SoS model was employed to simulate cell health within aggregates of varying sizes and extracellular metabolite concentrations. These simulation results were able to reproduce the trend observed for experimental findings. Specifically, the model predicted that larger iPSC aggregates would experience hypoxic conditions and poor nutrient supply for inner cells. This stress would then lead to reduced cell growth and increased cell necrosis at the center of the aggregates. Second, the Bio-SoS model was used to investigate the impact of aggregate size on both cell metabolism and micro-environmental heterogeneities. Those simulations provided quantitative insights into the variation of metabolic fluxes across cells at different positions within iPSC aggregates and under varying culture conditions. Third, the Bio-SoS model was used to identify the optimal aggregate size range to maximize the efficiency and yield of pluripotent stem cells cultured in a bioreactor. These simulations suggested an ideal aggregate size range is around 150 μm in radius for biomass productivity, which agreed with published reports[8,38].

The system of modules can be (a) assembled to facilitate data integration and improve the prediction of monolayer and aggregate cultures, and (b) utilized to control cell product quality heterogeneity and optimize the production process performance. Also, the modular design of the Bio-SoS model will facilitate future extensions to incorporate cell responses (i.e., flux rates, phenotype, and gene expression) to both mechanical (e.g., stirred speed and hydrodynamic force) and chemical (e.g., concentrations of nutrients/metabolites/oxygen, and pH) environmental conditions.

## Methods
**Cell proliferation and aggregation dynamics modeling.** The aggregation process is characterized by the dynamic evolution of a density profile of aggregates. Let $\phi(x, t)$ denote the average number of aggregates or clusters of size $x$ (i.e., mass and volume) at time $t$, where mass is equal to a buoyant density times volume. Due to the fact that buoyant density of cells does not vary significantly during the cell cycle, it was assumed the size $x$ corresponds either to the volume or mass having the relationship with radius (denoted by $R$)[14], i.e., $x \propto R^3$. The important factors impacting on the merge rate of cell aggregates include: (1) mass of aggregates; (2) position or distance of aggregates; and (3) velocity. Therefore, for any cell aggregate with mass $x$, the rate at which it merges with another aggregate with mass $x'$ is proportional to their densities, i.e., the average number of coalescences, $(x, x') \to x + x'$, per unit time per unit volume is $\frac{1}{2} \phi(x, t) \phi(x', t) K(x|x')$.

The aggregation kernel, denoted by $K(x|x')$, is associated with the average coalescence or merge rate of cell aggregates with mass $x$ and $x'$. In this work, only a purely coalescing process was considered where the merge rate was only dependent on the size of aggregates,

$$K(x|x') = k \cdot \exp\left(-k_1\left(\frac{x+x'}{2}\right)^a\right)\left(x^{\frac{1}{3}} + x'^{\frac{1}{3}}\right)^{\frac{7}{3}} \text{with constants } k, k_1, a > 0.$$
(1)

The exponential part in equation (1) accounts for the fact that there is a decreasing coalescence efficiency as the aggregate size increases[14]. The part $\left(x^{\frac{1}{3}} + x'^{\frac{1}{3}}\right)^{\frac{7}{3}}$ accounts for the fact that the collisions of these aggregates and the shear induced by micro-fluids with nonlinear velocity profile through a pore network within each cell cluster can lead to film drainage. It involves the draining of the film surrounding the interactive aggregates to permit actual coalescence of aggregates with size $x$ and $x'$. The effects from other factors are lumped into the constant $k$ of the model. In general, except aggregate size, the merge rate should be a function of other factors, such as agitation speed and medium compositions. The cell-to-cell and cell-to-medium interactions influence the aggregation process and the design of the kernel $K(\cdot)$. This falls outside the scope of this paper but is a subject for future work. The kernel parameters ($k = 1.26 \times 10^{-3} \text{ h}^{-1} \text{ μm}^{-7/3}$, $k_1 = 1.94 \times 10^{-4} \text{μm}^{-3a}$, $a = 8.06 \times 10^{-1}$) are adapted from Wu et al.[14].

The break-up effects of aggregates were considered negligible and the profile $\phi(x, t)$ evolves through coalescence events only. A PBM is employed to simulate the temporal evolution of size of the aggregates accounting for cell proliferation contributions and collisions between particles to aggregate size growth[14,41,42]. The temporal evolution of cell aggregate profile becomes,

$$\frac{\partial \phi(x,t)}{\partial t} = \frac{1}{2}\int_{x_0}^{x} \phi(x_C, t)\phi(x', t)K(x_C|x')dx'$$
$$- \int_{x_0}^{\infty}\phi(x,t)\phi(x',t)K(x|x')dx' - \frac{\partial\left[\phi(x,t)\frac{\partial x}{\partial t}\right]}{\partial x},$$
(2)

where the density function $\phi(x, t)$ is defined such that $\phi(x, t)dx$ is the fraction of aggregates with sizes between $x$ and $x + dx$ at time $t$ per unit volume of the culture. Let $x_0$ denote the size of a single cell. The first and second terms on the right side of equation (2) coming from the Smoluchowski coagulation equation, accounting for random coalescence. The first term describes the formation of aggregates with size $x'$ due to the aggregation of cell clusters with size $x' > x_0$ and $x_C = x - x'$. The last two terms represent the "loss" of aggregate with size $x$ due to their merging with those of size $x'$ to form larger aggregates and due to cell proliferation $\partial x/\partial t$.

Following Wu et al.[14], the rate of aggregate size was modeled by the change $\partial x/\partial t$ due to cell proliferation by Gompertz equation[43]:

$$\frac{\partial x}{\partial t} = \alpha_G \cdot x \log\left(\frac{M}{x}\right)$$

where $M$ is the aggregate size reached as $t \to \infty$ and $\alpha_G$ is a constant characterizing the cell proliferation. The Gompertz equation parameters ($M = 9.71 \times 10^6$ and $\alpha_G = 5.72 \times 10^{-3} \text{ h}^{-1}$) are adapted from Wu et al.[14]. Based on the iPSC culture data utilized in this study, the shrinking of iPSC aggregates has not been observed. If the experimental observation indicates the cell death impact, our model can be extended to account for the shrinking of the iPSC aggregates due to the cell death.

The transformation of the density function from aggregate size $x$ to aggregate radius $R$ can be done with the rule of

transformation of random variables. For the strictly monotone transformation $f : R \mapsto x^{\frac{1}{3}}$ and its inverse $x = f^{-1}(R) = R^3$ the probability density function of the aggregate radius, denoted by $\phi_R(R, t)$, is given by,

$$\phi_R(R, t) = \phi(f^{-1}(R))\left|\frac{df^{-1}(R)}{dR}\right| = 2\phi(R^3, t)R^2.$$

The first moments of the cell distribution yield the total aggregate size (accounting for the aggregate porosity $\varepsilon$), $M_1(t) = (1 - \varepsilon)\int_{x_0}^{\infty} x\phi(x, t)dx$. Since the size of each cell is $x_0$, the average number of cells in each aggregate is given by $z_{cell} = M_1(t)/x_0$. Let $X$ denote the cell density and $V$ is the volume. We can further calculate the total number of aggregates as $z_{total} = XV/z_{cell}$ and aggregate density (aggregate count per unit of volume) as $z_{density} = X/z_{cell}$.

Because the system comprises aggregates of multiple sizes, cell aggregates were classified into $L$ groups. The radius of each $\ell$th group with $\ell = 1, 2, \ldots, L$ is between $R_{\ell-1}$ and $R_\ell$. Remember that $R_0$ is the radius of a single cell. Thus, the cell aggregate density in the $\ell$th group is,

$$M_\ell(t) = z_{density}\int_{R_{\ell-1}}^{R_\ell} \phi_R(R, t)dR, \ell = 1, 2, \ldots, L.$$
(3)

**Reaction–diffusion model**. The RDM provides a means of characterizing cell-to-cell interactions and quantifying the spatial heterogeneity in micro-environmental conditions, by modeling the dynamic change of spatial and temporal concentration profiles of crucial components such as nutrients and metabolites like glucose and lactate. Suppose that the aggregates have cell spheres. A cluster of cells in a liquid medium was considered and assumed to have uniform diffusion in all directions along the radial axis. Let $s = \left(s_1, \ldots, s_{n_s}\right)$ denote the spatial profile of intracellular metabolite concentrations for species $i = 1, 2, \ldots, n_s$ at time $t$ and located at radius $r$. Let $c = \left(c_1, \ldots, c_{n_s}\right)$ denote the spatial profile of concentration of each extracellular species $i = 1, 2, \ldots, n_c$ at time $t$ and located at radius $r$.

The concentration of the $i$th species at time $t$ and located at aggregate radius $r$, denoted by $c_i(r, t)$, satisfies the set of reaction-diffusion equations in spherical coordinate with boundary conditions, i.e.,

$$\frac{\partial c_i}{\partial t} = \frac{D_i}{r^2}\frac{\partial}{\partial r}\left(r^2\frac{\partial c_i}{\partial r}\right) + \rho_i(c, s)$$
(4)

subject to boundary conditions : $(a)\, c_i(R, t) = u_i(t)$ and $(b)\, \frac{\partial c(0, t)}{\partial r} = 0$

where $D_i$ is the effective diffusion coefficient, $u_i(t)$ is the extra-aggregate environmental conditions (e.g., the bulk concentrations of glucose and lactate in the bioreactor), and $R$ is the aggregate size. Here the reaction rate $\rho_i$ is negative for nutrient consumption and positive for inhibitor formulation. The boundary condition (a) comes from the fact that the metabolite concentration on the surface of an aggregate equals to that measured in the bioreactor; and (b) is produced by the condition of spherical symmetry.

Due to the reaction and diffusion, the metabolite concentrations at different locations or depth of a cell aggregate are different. We divide each aggregate of radius $R_\ell$ into $N_\ell$ spherical shells (shaped like a 3D annulus or "rings") and assume that the cellular metabolisms are homogeneous in each spherical shell. The $n$th spherical shell of the $\ell$th aggregate is located between the radii between $R_\ell^{n-1}$ and $R_\ell^n$; see the illustration in Fig. 1b. The

volume of the spherical shell is the difference between the enclosed volume of the outer sphere and the enclosed volume of the inner sphere: $\frac{4\pi}{3}\left(R_\ell^{(n)}\right)^3 - \frac{4\pi}{3}\left(R_\ell^{(n-1)}\right)^3$.

*Quasi-steady-state solution in radial cases of $c_i(r,t)$.* It is challenging to directly solve the transient reaction-diffusion equation (4). Thus, a similar approach as that used in McMurtrey[44] was applied to account for the quasi-steady-state setting, i.e., $\frac{\partial c_i}{\partial t} = 0$. A quasi-steady-state solution of RDM is widely used in cell aggregation literature[44,45] to describe the metabolite diffusion inside an aggregate. In each $n$th spherical shell, the extracellular concentration profile or micro-environmental condition can be solved analytically[44] as the nutrient consumption or inhibitor formulation rates $\rho_i(c,s)$ are constant in the spherical shell specified by $[R_\ell^{n-1}, R_\ell^n]$:

$$c_i(r,t) = \frac{1}{6}\rho_i^{(\ell,n,t)}D_i\left(R_\ell^{(n)2} - r^2\right) + u_i^{(\ell,n,t)} \text{ with } r \in \left[R_\ell^{n-1}, R_\ell^n\right] \tag{5}$$

where $\rho_i^{(\ell,n,t)}$ is the consumption/formulation rate of metabolite $W_i$ in the $n$th spherical shell of the $\ell$th aggregate and $u_i^{(\ell,n,t)}$ denotes the the boundary condition of metabolite $W_i$ in the $n$th spherical shell of the $\ell$th aggregate at time $t$. The formulation/consumption rate $\rho_i^{(\ell,n,t)}$ depends on the intracellular metabolism of cells located in the $n$th spherical shell of the $\ell$th aggregate and its mathematical formulation will be discussed in the next section.

Here, $D_i$ is the effective diffusion coefficient of the $i$th species. Given the porosity $\varepsilon$ and tortuosity $\tau$, it is calculated as $D_i = \frac{\varepsilon}{\tau} \cdot D_i^a$, where $D_i^a$ is the diffusion coefficient of the $i$th species in aqueous condition. In a 4-day culture in spinner flasks at the agitation rate of 60 rpm, the porosity and tortuosity of hESC aggregates were reported to be $0.270 \pm 0.007$ unitless and $1.551 \pm 0.086$ unitless, respectively[14]. Since the metabolites are transported between the outer spherical shell and extra-aggregate environment, the boundary condition for the $N_\ell$th spherical shell is given by $u_i^{(\ell,N_\ell,t)} = u_i(t)$. Further, metabolites are freely transported between spherical shells and thus the metabolite concentrations on the outer surface of the $(n-1)$th spherical shell are equal to the concentrations on the inner surface of the $n$th spherical shell, i.e., mathematically, $u_i^{(\ell,n-1,t)} = c_i\left(R_\ell^{(n)}, t\right)$. The solution in equation (5) illustrates that: (a) the nutrient concentrations, increasing from aggregate center to the surface, become highest on the surface $c(R,t) = u_i(t)$; and (b) the metabolite waste concentrations, decreasing from center to surface, is highest at the center

$$c(0,t) = u_i(t) + \frac{1}{6}\sum_{n=1}^{N_\ell}\rho_i^{(\ell,n,t)}D_i\left(\left(R_\ell^{(n)}\right)^2 - \left(R_\ell^{(n-1)}\right)^2\right).$$

*Estimating reaction rate $\rho_i$ in RDM.* Conceptually the reaction rate $\rho_i$ (nmol/($\mu m^3 \cdot h$)) of the metabolite $W_i$ is the weighted sum of the associated flux rates (nmol/$10^6$ cells/h). Thus, putting everything in vectors, we have $\rho = N \cdot v \cdot \gamma$ where $N$ represents the stoichiometric matrix, $v$ is the flux rate vector, and $\gamma$ is a unit conversion factor. By assuming the averaged single-cell radius to be 7.5 $\mu m$, the average cell volume can be estimated by $A = \varepsilon \cdot \frac{4\pi}{3} \times 7.5^3$ ($\mu m^3$/cell) with porosity $\varepsilon$. Then 1 nmol/$10^6$ cells/h of flux rate can be converted to the reaction rate of

$$\gamma = \frac{1 \text{ nmol}/10^6 \text{ cells/h}}{A} = \frac{3 \times 10^6}{4\pi\varepsilon \times 7.5^3}\mu m^3 \cdot h.$$

**Stochastic metabolic reaction network model for single-cell.** For each iPSC, let us consider a metabolic network system with $n$

species $(W_1, W_2, \ldots, W_n)$ which interact with each other through $k$ chemical reactions. Let $\widetilde{u}_i(t)$ denote the number of molecules of metabolite $W_i$ in an individual cell at time $t$. Denote the vector $\widetilde{u} = (\widetilde{u}_1, \widetilde{u}_2, \ldots, \widetilde{u}_n)$. Here $\widetilde{u}$ includes both intracellular and extracellular metabolites, i.e., $\widetilde{u} = (\widetilde{s}, \widetilde{c})$. The metabolic reaction network can be expressed as[46]

$$\sum_{i=1}^{n}\eta_{ij}W_i \xrightarrow{v_j} \sum_{i=1}^{n}\eta'_{ij}W_i, j = 1, 2, \ldots, k \tag{6}$$

where $\eta_{ij}$ and $\eta'_{ij}$ are nonnegative integers. Let $\boldsymbol{\eta}_j$ be the vector whose $i$th component is $\eta_{ij}$ representing the number of molecules of the $i$th metabolite consumed in the $j$th reaction. Let $\boldsymbol{\eta}'_j$ be the vector whose $i$th component is $\eta'_{ij}$ representing the number of molecules of the $i$th metabolite produced by the $j$th reaction. By writing them in matrix form, i.e., $\boldsymbol{\eta} = (\boldsymbol{\eta}_1, \ldots, \boldsymbol{\eta}_k)$ and $\boldsymbol{\eta}' = (\boldsymbol{\eta}'_1, \ldots, \boldsymbol{\eta}'_k)$, we define the stoichiometric matrix as $N = \boldsymbol{\eta}' - \boldsymbol{\eta}$. Let $v_j(\widetilde{u})$ be the flux rate at which the $j$th reaction occurs for an individual cell, that is, the propensity/intensity of the $j$th reaction as a function of the number of molecules of metabolites. Later on, we will show that the rate $v_j(\widetilde{u})$ is a generalized flux rate, analogous to that of the MFA.

Let $R(t)$ be the $k$-dimensional vector whose $j$th component is $R_j(t)$, representing the number of times the $j$th molecular reaction has occurred by time $t$ in a single cell. Thus, at time $t$, the profile of intracellular and extracellular metabolite molecules follows the dynamics, i.e.,

$$\widetilde{u}(t) = \widetilde{u}(0) + \sum_{j=1}^{k}R_j(t)\left(\boldsymbol{\eta}'_j - \boldsymbol{\eta}_j\right) = \widetilde{u}(0) + N \cdot R(t). \tag{7}$$

Equation (7) is a mass balance equation where $\widetilde{u}(t) - \widetilde{u}(0)$ is the difference of a number of molecules in time interval $[0, t]$ and $N \cdot R(t)$ is the net amount of reaction output by time $t$. The number of occurrences of the $j$th molecular reaction, $R_j(t + dt) - R_j(t)$, during time interval $[t, t + dt]$ is modeled as a Poisson random variable with mean (and variance) $v_j(\widetilde{u}(t))dt$ and $R_j(t)$ follows a nonhomogeneous Poisson process with the flux rate or molecule generation rate $v_j(\widetilde{u}(t))$[17,20,46].

Based on the definition of Poisson process, during the time interval $(t, t + dt]$, the probability that the $j$th reaction occurs $n$ times becomes[46,47]

$$\begin{aligned}&P\left(R_j(t + dt) - R_j(t) = n\right)\\&= \frac{e^{-\int_t^{t+dt}v_j(\widetilde{u}(x))dx}\left(\int_t^{t+dt}v_j(\widetilde{u}(x))dx\right)^n}{n!} \overset{\text{def}}{=} \text{Poisson}\left(\int_t^{t+dt}v_j(\widetilde{u}(\tau))d\tau\right)\end{aligned} \tag{8}$$

Define the expected accumulated occurrences of the $j$th molecular reaction by time $t$, i.e., $\Lambda_j(t) = \int_0^t v_j(\widetilde{u}(\tau))d\tau$. Equation (8) can be rewritten as $R_j(t) = Y\left(\Lambda_j(t)\right)$, where $Y(t)$ is a unit (or rate one) Poisson process. Then equation (7) becomes,

$$\widetilde{u}(t) = \widetilde{u}(0) + \sum_{j=1}^{k}Y\left(\Lambda_j(t)\right)\left(\boldsymbol{\eta}'_j - \boldsymbol{\eta}_j\right). \tag{9}$$

**Bio-SoS mechanistic model for iPSC aggregate culture.** By assembling the single-cell metabolic network with population balance and RDMs, we developed the Bio-SoS model characterizing the dynamics of iPSC aggregate culture with either homogeneous or non-homogeneous cell population. Since cells create the driving force for the dynamics of the culture process, we focus on modeling the evolution of cell metabolic dynamics and associated microenvironmental conditions. Let $X$ denote the cell density, i.e., number of cells per unit of volume. Let $\widetilde{u}_b(t)$ denote

the number of molecules of metabolites within and "around" each $b$th cell with $b = 1, 2, \ldots, X$.

First, the homogeneous cell population is assumed. Let $\boldsymbol{u}(t, X) = \sum_{b=1}^{X} \widetilde{\boldsymbol{u}}_b(t)$ denote metabolite concentrations (i.e., the number of molecules of metabolites per unit of volume) in the system at time $t$. Given a cell density $X(t)$ (i.e., the number of cells per unit of volume), the change in the number of metabolite molecules during the short time interval $(t, t + dt]$ is obtained by summing the changes from each cell, i.e.,

$$\Delta \boldsymbol{u}(t, X) \equiv \boldsymbol{u}(t + \Delta t, X) - \boldsymbol{u}(t, X) = \sum_{b=1}^{X} \left[ \widetilde{\boldsymbol{u}}_b(t + \Delta t) - \widetilde{\boldsymbol{u}}_b(t) \right]$$

$$= \sum_{b=1}^{X} \sum_{j=1}^{k} Y \left( \int_{t}^{t+\Delta t} v_j(\widetilde{\boldsymbol{u}}_b(\tau)) d\tau \right) \left( \boldsymbol{\eta}'_j - \boldsymbol{\eta}_j \right)$$

$$= \sum_{j=1}^{k} Y \left( X \int_{t}^{t+\Delta t} v_j(\widetilde{\boldsymbol{u}}_b(\tau)) d\tau \right) \left( \boldsymbol{\eta}'_j - \boldsymbol{\eta}_j \right)$$

(10)

When the flux rate is constant $v_j(\widetilde{\boldsymbol{u}}_b(t)) = v_j$, the dynamics of expected metabolite concentration are the same as the dynamic flux balance model[15,48] (see "Supplementary Methods"). Thus, the deterministic ODE-based metabolic model can be interpreted as a special case of the stochastic reaction model (Eq. (10)) with mean metabolite concentrations and constant flux rates, ignoring cell-to-cell variation.

Second, the heterogeneous cell population is considered. Let $u_i(t)$ denote the concentration of metabolite $W_i$ (i.e., the number of molecules per unit of volume) in the system at time $t$. Here the metabolite concentrations include both intracellular and extracellular metabolites, i.e., $\boldsymbol{u} = (\boldsymbol{s}, \boldsymbol{c})$. Recall that the cell aggregates were divided to $L$ groups with different size. Each aggregate is divided to $N_\ell$ spherical shells with $\boldsymbol{c}(R_\ell^n, t)$ and $\boldsymbol{s}(R_\ell^n, t)$ representing the concentrations of extracellular (within aggregate) and intracellular metabolites in the $\ell$th aggregate at the radius of $R_\ell^n$ at time $t$. Thus, the cell density (i.e., the cell number per unit of volume) in the $n$th spherical shell of the $\ell$th aggregate group at time $t$ can be computed based on the aggregate density and the number of cells in that spherical shell, i.e.,

$$G_\ell^{(n)}(t) = M_\ell(t)(1 - \varepsilon) \frac{\left( R_\ell^{(n)} \right)^3 - \left( R_\ell^{(n-1)} \right)^3}{R_0^3}$$

where the aggregate density $M_\ell(t)$ (Eq. (3)) was obtained by solving the PBM.

Then, for each $j$th reaction, the overall flux rate of all cells in a unit of volume residing in the spherical shell $\left[ R_\ell^{(n-1)}, R_\ell^{(n)} \right]$ at time $t$ (denoted by $\bar{v}_j^{(\ell, n, t)}$) is given by the product of the cell density $G_\ell^{(n)}$ and single-cell flux rate $v_j^{(\ell, n, t)} = v_j \left( \boldsymbol{c}\left( R_\ell^{(n)}, t \right), \boldsymbol{s}\left( R_\ell^{(n)}, t \right) \right)$ in the shell, i.e.,

$$\bar{v}_j^{(\ell, n, t)} = G_\ell^{(n)}(t) v_j^{(\ell, n, t)}.$$

Recall that $c\left( R_\ell^{(n)}, t \right)$ can be obtained by solving the RDM (see "Methods" section).

After that, we can divide the culture time $[0, t]$ into $G$ equally spaced intervals $\left[ (g-1)\Delta t, g\Delta t \right]$ with $g = 1, 2, \ldots, G$, and numerically compute the metabolite concentrations $\boldsymbol{u}(t)$ as

$$\boldsymbol{u}(t) \approx \boldsymbol{u}(0) + \sum_{\ell=1}^{L} \sum_{n=1}^{N_\ell} \sum_{g=1}^{G} \sum_{j=1}^{k} Y \left( \bar{v}_j^{(\ell, n, (g-1)\Delta t)} \Delta t \right) \left( \boldsymbol{\eta}'_j - \boldsymbol{\eta}_j \right)$$

where $\boldsymbol{u}(0)$ is the initial concentrations.

**Variance decomposition analysis.** Stem cells, such as iPS cells that proliferate in the culture process can undergo considerable

variation in metabolic reaction rates due to the change in specific micro-environmental conditions. To better understand the metabolic heterogeneity of stem cell culture and identify sources of uncertainty, we developed a variance decomposition method based on the Bio-SoS model. It can be used to explain how the variance measuring the metabolic heterogeneity is contributed by different-sized cell aggregates and how the variance is spatially distributed within each cell aggregate.

- Single aggregate: Let $\sum_{j=1}^{k} Y(\bar{v}_j^{(\ell, n, t)} \Delta t)(\eta'_{ij} - \eta_{ij})$ represent the concentration change of metabolite $W_i$ due to the reactions occurring in cells located in the $n$th spherical shell of the $\ell$th aggregate during the time interval $(t, t + dt]$. Then, the total variance of the metabolite $W_i$ in an individual aggregate, denoted by $\sigma_{i, \ell, t}^2$, can be expressed by

$$\sigma_{i, \ell, t}^2 = Var\left( \sum_{n=1}^{N_\ell} \Delta u_i^{(\ell, n, t)} \right) = Var\left[ \sum_{n=1}^{N_\ell} \sum_{j=1}^{k} Y\left( \bar{v}_j^{(\ell, n, t)} \right) \left( \eta'_{ij} - \eta_{ij} \right) \right].$$

RSD is a statistical measurement that describes the spread of data with respect to the mean. The RSD of metabolite $W_i$ in each $\ell$th cell aggregate at time $t$ is expressed by

$$\text{RSD}_{i, \ell, t} = \frac{\sigma_{i, \ell, t}}{\mu_{i, \ell, t}} \text{ with } \mu_{i, \ell, t} = \mathbb{E}\left[ \sum_{n=1}^{N_\ell} \Delta u_i^{(\ell, n, t)} \right]. \quad (11)$$

- Cell population heterogeneity: Given well-controlled bulk bioreactor conditions, we suppose the independence of different aggregates. During the time interval $(t, t + dt]$, the total variance of the metabolite $W_i$ concentration change in the bio-system, denoted by $\sigma_{i, t}^2$, can be divided into the contribution from each $\ell$th group of aggregates with radius $R_\ell$, i.e.,

$$\sigma_{i, t}^2 = Var\left[ \sum_{\ell=1}^{L} \sum_{n=1}^{N_\ell} \sum_{j=1}^{k} Y\left( \bar{v}_j^{(\ell, n, t)} \Delta t \right) \left( \eta'_{ij} - \eta_{ij} \right) \right] = \sum_{\ell=1}^{L} \sigma_{i, \ell, t}^2.$$

This study can support the analysis and provide insights into both metabolic heterogeneity and spatial heterogeneity. The metabolic heterogeneity can be assessed by studying the relative changes of metabolic fluxes for cells residing at different locations and times within an aggregate, as shown in Fig. 7. At the same time, the impact of spatial heterogeneity can be investigated by measuring the contribution of (biomass) output variance from the cell aggregates with different radius size as illustrated in Fig. 8. This can guide the selection of optimal aggregate size to maximize the expected yield and control the output variance.

**Flux and metabolite standardization.** Standardization of the flux rates and metabolite concentrations used was performed in three steps: (1) the average fluxes and metabolite concentrations were calculated for each $\ell$th group of aggregates with radius $R_\ell$ ranging from 30 to 600 μm; (2) the means and standard deviations of those fluxes and concentrations over all aggregates were computed; and (3) for each flux or metabolite concentration, the mean value was subtracted and the result was divided by the corresponding standard deviation. For example, the average flux of the $j$th reaction in the $\ell$th group of aggregates is given by $\bar{v}_j^{(\ell, \cdot, t)} = \frac{1}{N_\ell} \sum_{n=1}^{N_\ell} \bar{v}_j^{(\ell, n, t)}$ and the standardization of this within-aggregate flux is performed by

$$\text{Standardization}\left( \bar{v}_j^{(\ell, \cdot, t)} \right) = \frac{\bar{v}_j^{(\ell, \cdot, t)} - \bar{v}_j^{(\cdot, \cdot, t)}}{\sqrt{\frac{\sum_{\ell=1}^{L} \left( \bar{v}_j^{(\ell, \cdot, t)} - \bar{v}_j^{(\cdot, \cdot, t)} \right)^2}{L-1}}}$$

where $\bar{v}_j^{(\cdot, \cdot, t)} = \frac{1}{L} \sum_{\ell=1}^{L} \bar{v}_j^{(\ell, \cdot, t)}$.

**Statistics and reproducibility**. The details about sample sizes, parameters, and steps of statistical analysis are provided in relevant methods and results sections, figure legends, and tables where applicable. All statistical analysis is performed in Python.

**Reporting summary**. Further information on research design is available in the Nature Portfolio Reporting Summary linked to this article.

## Data availability
The data used in this work are collected from open-access databases. The monolayer iPSC cell culture data are in Supplementary Data[11]. The bioreactor iPSC culture data are from Kwok et al.[7]. The parameters of the population balance model are adapted from Wu et al.[14]. The diffusion coefficients are from multiple published sources[49–55].

## Code availability
The main results are generated using Python 3. The code generating the main results in this work is available at the GitHub repository. A supplementary file (zip file) containing the code used in the paper, associated test data, parameters, and documentation is available.

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

## Acknowledgements

We gratefully acknowledge funding support from the National Institute of Standards and Technology (Grant nos. 70NANB17H002 and 70NANB21H086) to Dr. Wei Xei and the National Science Foundation (Grant nos. OIA-1736123 and EEC-2100442) to Dr. Sarah W. Harcum.

## Author contributions

H.Z. developed the model, designed the data analyses, visualized the results, and wrote the original draft. S.W.H. supervised the study, provided the iPSC monolayer culture data, and analyzed and interpreted the results. J.P. contributed to the introduction session and reviewed the entire paper. W.X. proposed the idea of the biological system-of-systems framework for iPSC aggregate culture, supervised the methodology development, assessed the model performance, and reviewed and improved the paper. All authors have read and approved the final paper, contributing edits where applicable.

## Competing interests

The authors declare no competing interests.
