## [Peer Review File · Communications Biology]

Reviewers' comments:

Reviewer #1 (Remarks to the Author):

The goal of the authors is to address the challenges of large-scale manufacturing of induced pluripotent stem cells (iPSCs) for cell therapies and regenerative medicines. To tackle these challenges, authors propose a novel Biological System-of-Systems (Bio-SoS) framework to characterize cell-to-cell interactions, spatial heterogeneity, and cell response to micro-environmental variation. This Bio-SoS framework introduces a new approach to multi-scale bioprocess mechanistic modeling. It focuses on iPSC aggregate cultures, analyzing metabolic dynamics, spatial heterogeneity, and interactions over time. The model is versatile, studying cell health, predicting metabolite concentrations, and optimizing aggregate size and feeding strategies for bioreactors. Validation shows its transferability across cell lines and conditions. It's applied in three scenarios: simulating cell health in various conditions, exploring aggregate size effects, and finding optimal sizes for stem cell production. The modular design allows future extensions like hydrodynamics and gene regulatory networks, improving iPSC culture predictions for control and quality consistency.

Overall, this paper is well-written, nicely presented and makes a valuable contribution to the field. The content aligns well with the focus of the journal and I recommend its publication. However, prior to final acceptance, I suggest addressing minor concerns to enhance clarity and precision in certain sections of the manuscript.

Minor comments:

- It seems that figure numbering went wrong when building the pdf, as figures start at Figure 5. The .docx document looks OK though.
- In Figure 3. The description associate "blue" lines with the mechanistic model from Wang et al, whereas in the legend those lines are "green"
- Supplementary figure numbering seems to be wrong in the .docx file too (e.g. Supplementary Fig. 9 is referenced at line 262)
- The biomass production presents a sharp slowdown at 72h, which suggests the depletion of nutrients, but it is unclear how smooth this transition is with respect to the 48h mark. Perhaps it would be useful to simulate more intermediate time points and show the reader the corresponding biomass production using the same y-axis scale (or even as a 3D surface with time as the third axis).
- The discussion should highlight more the novelty of the work, especially considering that the authors label it as a "revolutionary step". The specific capabilities provided by the framework that didn't exist previously or that have been greatly enhanced should be discussed in detail. I'm also missing some explanation on how the average user, not necessarily versed with coding, can make use of the framework. This should be discussed.
- Authors state that the python code was included in the .zip file for review. It was not. I would appreciate to have it available when the authors reply.

Reviewer #2 (Remarks to the Author):

This article discussed the development of a framework for considering 3-dimensional multiscale models of induced pluripotent stem cell (iPSC) aggregate dynamics. The authors highlight the complexity of these multiscale processes due to large spatial heterogeneities across a large aggregate system. They introduce a modular framework that incorporates stochastic models of reaction networks, a reaction-diffusion model for the bioreactor conditions, and a population balance model for the cell population. The framework is composed of Stochastic Partial Differential Equations (SPDEs) that describe the evolution of the mass/volume density of cell aggregates over space and time. This work only considers the coalescence of aggregates merging at a size dependent rate (not splitting or non-spherical shape changes) based on an empirical model function. The stochasticity is implemented as Poissonian distributed reaction events within annuli areas of the aggregate. However, it is unclear what the main contribution of this article is. It appears that the reaction modelling of a companion paper (reference 21 in the article) has been coupled to a spatially dependent reaction term of a reaction-diffusion PDE where the spatial dependency is determined by a second PDE representing aggregate growth. The simulation results were compared to experimental results (also from reference 21). The main contribution could be made more explicit.

The authors motivate the need for this framework due to the need to develop modelling paradigms to inform clinical work. In particular, the authors claim that "... a comprehensive model that captures the multi-scale and heterogeneity nature of iPSC aggregate cultures is still lacking". Multiscale models for similar aggregate structures (mammalian aggregates, cancer tissue studies, biofilms) have been made, (see PhysiCell, Chaste, ChemChaste, BMX, Morpheus). Under the umbrella term of agent-based models, these software packages couple cell-based metabolic reaction networks with reaction-diffusion and individual cell growth dynamics. Additionally, these models can incorporate specialized stochastic cell models to represent regulatory structures and cell-cell physical dynamics through calculating the forces acting on a cell. While strong arguments can be made to forego such computational complexity in models for iPSC aggregates, to not mention these agent-based models seems like an oversight. Specifically, in this case where the authors propose a multi-scale 'mechanistic' model considering processes at the individual cell, aggregate, and cell population levels; all aspects handled by the agent-based software packages. This potential gap in the narrative is only amplified when the authors write "... the Bio-SoS framework marks a revolutionary step in the mechanistic model development of multi-scale bioprocess...", albeit written in the discussion. This statement is somewhat hyperbolic as there is little evidence to suggest that the proposed framework "effectively characterizing cell-to-cell interactions and complex mechanisms of iPSC aggregate culture" as no physical interactions are considered.

In contrast the authors model aggregates as a series of annuli that assumes each cell across the aggregate annulus has the same phenotype and behavior despite the discussed heterogeneous conditions. The authors demonstrate that the stochastic model may be reduced to a PDE model with metabolic flux analysis representing the reaction terms. The authors also only average over 10 simulation outputs. Therefore, the evidence to support the need to consider stochastic models is weak and the error due spatial assumptions may override a stochastic signal. For the concentrations of the metabolites found in a cell, is a stochastic mathematical model necessary? Are the authors predicting a phenomenon that could not be achieved through deterministic models of chemical reactions? These aspects would need to be discussed to provide support to the framework. This framework may find some success as a coarse-grained simulation of aggregate structures. The simulation time will be quicker than more complex models and may be more suitable for an industrial application.

Specific comments:

1. The framework code was not provided.

2. Further proof reading by the authors is recommended (see annotated .pdf).

3. Line 194, "which provided an excellent opportunity to predict iPSC culture behavior not used in the training of the model." It's not clear what this training entails, were core model parameters optimized for and applied to future model experiments?

4. The excretion of extracellular matrix materials was mentioned in the introduction, but the cell/aggregate models do not exhibit such a capability.

5. Similarly, how was the 'validation' for the model from the data performed? Statistical or qualitative validation?

6. The metabolic model only considers enzymatic reactions in a kinetic manner, using Monod kinetics. The biomass growth flux objective is the product of the amino acid synthesis and does not consider other objective products such as cell wall or protein synthesis as used in more complex biomass objectives. This metabolic model does not account for enzyme cost or higher control structures of the cell. A comment on the production of the biomass objective would be appreciated.

7. Figure 6 is difficult to read

8. Pages 13 and 14, which figure is denoted as 'Figure 11' is unclear. The figures and captions look muddled, line 312 looks like a caption?

9. A finite difference of grid space 15 micrometer and aggregate radius in steps of 60 micrometer, leaving only 4 finite grid difference. Is this sufficient? The diameter of a cell is also 15 micrometer. The optimal aggregate was seen to be 10 grid cells in radius, is this step size precise enough for accurate simulation? What is the temporal step size to ensure numerical stability (Courant-Friedrichs-Lewy condition) of PDEs?

10. Lines 174-177. The analysis presented ties the aggregate dynamics as a balance between resource consumption in the interior and buildup of metabolic waste products. In both experiments (Liu, J. et al. (2015).) and models (Bocci, F. et al. (2018).) on the similar biofilm system, this same interplay was seen to induce film/aggregate radius oscillations, was this seen in the model or can the authors comment on whether oscillating dynamics were/could be seen in the aggregate model? Shrinking of the aggregate would require cell death which, while mentioned, does not seem to be included in the growth model.

References

Liu, J. et al. (2015). Metabolic co-dependence gives rise to collective oscillations within biofilms. *Nature*, 523, 550–554.

Bocci, F. et al. (2018). Role of metabolic spatiotemporal dynamics in regulating biofilm colony expansion. *Proceedings of the National Academy of Sciences*, 115, 4288–4293.

Ghaffarizadeh A, Heiland R, Friedman SH, Mumenthaler SM, Macklin P (2018) PhysiCell: An open source physics-based cell simulator for 3-D multicellular systems. *PLOS Computational Biology* 14(2): e1005991. <https://doi.org/10.1371/journal.pcbi.1005991>

Johnson CGM, Fletcher AG, Soyer OS. ChemChaste: Simulating spatially inhomogeneous biochemical reaction-diffusion systems for modeling cell-environment feedbacks. *Gigascience*. 2022 Jun 17;11:giac051. doi: 10.1093/gigascience/giac051. PMID: 35715874; PMCID: PMC9205757.

F.R. Cooper, R.E Baker, M.O. Bernabeu, R. Bordas, L. Bowler, A. Bueno-Orovio, H.M. Byrne, V. Carapella, L. Cardone-Noott, J. Cooper, S. Dutta, B.D. Evans, A.G. Fletcher, J.A. Grogan, W. Guo, D.G. Harvey, M. Hendrix, D. Kay, J. Kursawe, P.K. Maini, B. McMillan, G.R. Mirams, J.M. Osborne, P. Pathmanathan, J.M. Pitt-Francis, M. Robinson, B. Rodriguez, R.J. Spiteri, D.J. Gavaghan. Chaste: Cancer, Heart and Soft Tissue Environment. *J. Open Source Softw.* 5(47):1848, 2020. doi: 10.21105/joss.01848.

Palmer, B.J., Almgren, A.S., Johnson, C.G.M. et al. BMX: Biological modelling and interface exchange. *Sci Rep* 13, 12235 (2023). <https://doi.org/10.1038/s41598-023-39150-1>

Starruß J, de Back W, Brusch L, Deutsch A. Morpheus: a user-friendly modeling environment for multiscale and multicellular systems biology. *Bioinformatics*. 2014 May 1;30(9):1331-2. doi: 10.1093/bioinformatics/btt772. Epub 2014 Jan 17. PMID: 24443380; PMCID: PMC3998129.

Reviewer #1

The goal of the authors is to address the challenges of large-scale manufacturing of induced pluripotent stem cells (iPSCs) for cell therapies and regenerative medicines. To tackle these challenges, authors propose a novel Biological System-of-Systems (Bio-SoS) framework to characterize cell-to-cell interactions, spatial heterogeneity, and cell response to micro-environmental variation. This Bio-SoS framework introduces a new approach to multi-scale bioprocess mechanistic modeling. It focuses on iPSC aggregate cultures, analyzing metabolic dynamics, spatial heterogeneity, and interactions over time. The model is versatile, studying cell health, predicting metabolite concentrations, and optimizing aggregate size and feeding strategies for bioreactors. Validation shows its transferability across cell lines and conditions. It's applied in three scenarios: simulating cell health in various conditions, exploring aggregate size effects, and finding optimal sizes for stem cell production. The modular design allows future extensions like hydrodynamics and gene regulatory networks, improving iPSC culture predictions for control and quality consistency.

Overall, this paper is well-written, nicely presented and makes a valuable contribution to the field. The content aligns well with the focus of the journal and I recommend its publication. However, prior to final acceptance, I suggest addressing minor concerns to enhance clarity and precision in certain sections of the manuscript.

Response: We appreciate the reviewer's feedback. In the revision, we improve the presentation and address the concerns.

Minor comments:

1. It seems that figure numbering went wrong when building the pdf, as figures start at Figure 5. The .docx document looks OK though.

Response: We appreciate for the comment. In the revision, we have corrected the issue.

2. In Figure 3. The description associate "blue" lines with the mechanistic model from Wang et al, whereas in the legend those lines are "green."

Response: Thanks for the comment. In the revision, we have corrected the typo and changed the description to be "green" lines to be consistent with the results in the plot.

3. Supplementary figure numbering seems to be wrong in the .docx file too (e.g. Supplementary Fig. 9 is referenced at line 262).

Response: Thanks for the comment. We have corrected the figure numbering issues.

4. The biomass production presents a sharp slowdown at 72h, which suggests the depletion of nutrients, but it is unclear how smooth this transition is with respect to the 48h mark.

Perhaps it would be useful to simulate more intermediate time points and show the reader the corresponding biomass production using the same y-axis scale (or even as a 3D surface with time as the third axis).

Response: We appreciate for the comment. In the revised paper, we added a series of biomass production result plots associated to 0, 6, 12, ..., 72 culture hour in **Supplementary Fig. 3**

See the change in Line 991-999 of Supplementary information.

5. The discussion should highlight more the novelty of the work, especially considering that the authors label it as a "revolutionary step". The specific capabilities provided by the framework that didn't exist previously or that have been greatly enhanced should be discussed in detail.

Response: We appreciate for the comment. In the revision, we improve the presentation and clearly describe the key contributions of this study. In specific, the novelty is summarized below.

- a) Our approach is a first attempt of developing a multiscale bioprocess mechanistic model, called the biological system-of-systems (Bio-SoS), for iPSC aggregate culture, characterizing the causal interdependencies at molecular, cellular, and macro-scope levels. It integrates intracellular, intra-aggregate, and medium bulk culture conditions. Given any feeding protocol and media compositions, it allows us to predict the production process outputs, i.e., yield and cell product quality consistency. Therefore, the Bio-SoS can facilitate the optimization of iPSC culture processes.
- b) The proposed Bio-SoS model has modular design. It includes three modules: (1) single-cell stochastic metabolic network model describing cell response to environmental variation; (2) population balance model characterizing cell-to-cell interaction and aggregation process; and (3) reaction-diffusion model characterizing spatial heterogeneity of micro-environments. This modular design facilitates assembling different production processes, which can accelerate the production process development for iPSC large-scale manufacturing without extensive experiments. Fig 4 illustrates that the Bio-SoS model can use 2D petri-dish monolayer culture data (that are often collected in the Lab) to provide reliable predictions of 3D aggregate cultures in in suspension bioreactors recommended for real industrial production. Instead of performing an extensive number of expensive and complex aggregate culture experiments in bioreactors and measuring the metabolite concentration within aggregates, the Bio-SoS model can be estimated using data from simpler, more cost-effective monolayer cultures in petri-dish.

Therefore, the system of modules can be a) assembled to facilitate data integration and improve the prediction of monolayer and aggregate cultures; and b) utilized to control cell product quality heterogeneity and optimize the production process performance. Also, the modular design of the Bio-SoS model will facilitate future extensions to incorporate cell responses (i.e., metabolic flux rates, phenotype, gene expression, pluripotency) to both

mechanical (e.g., stirred speed, hydrodynamic force, shear stress) and chemical (e.g., concentrations of nutrients/metabolites/oxygen, pH) environmental conditions.

- c) To enhance computational efficiency, a problem often encountered with other simulation software/tools, the simulation complexity of the Bio-SoS model does not increase as the manufacturing scale or bioreactor size increases due to various reasons, including: (1) a coarse-grained approach that divides each aggregate into small spherical shells and assumes that the cellular metabolisms are homogeneous in each spherical shell; (2) a single-cell stochastic metabolic network model that can characterize the cell metabolic response to environmental variation (i.e., environmental condition in different spherical shells); and (3) a probabilistic state dynamic model quantifying the number and proportion of cells in different discretized homogeneous environmental conditions.

In specific, cell aggregates were divided to L groups with different size. Each aggregate is divided to N_ℓ spherical shells. Consequently, there are $\left(\frac{1}{L} \sum_{\ell=1}^L N_\ell\right)$ number of partitions with each corresponding to specific homogeneous micro-environmental condition. We have a single-cell stochastic metabolic network model that can characterize cell response to different environmental conditions. This fixed number of partitions doesn't change with the bioreactor's size, ensuring that Bio-SoS scales effectively for large-scale iPSC industrial manufacturing. Consequently, our framework achieves a significant improvement in computational efficiency compared to the general multiscale simulation. The overall simulation time for a 72-hour iPSC culture of millions of cells is about 0.1 CPU hour.

- d) Since the spatial heterogeneity can contribute to the cell product quality variation, we derive a variance decomposition analysis on the Bio-SoS mechanistic model to quantify the contribution of aggregates with different size to the cell metabolic heterogeneity. The proposed variance decomposition analysis improves the understanding of spatial heterogeneity within iPSC aggregate culture and guides the control of aggregate size, which can avoid unwanted cell death and ensure product quality consistency during cell culture and expansion process.

In addition to methodological novelty, this paper also presents the key contributions for iPSC culture and cell processing, including that the Bio-SoS (1) has the potential to advance the understanding of underlying mechanisms; (2) predicts the impact of critical factors (i.e., aggregate size) on iPSC metabolic heterogeneity; and (3) provides a valuable tool for yield optimization and product quality consistency control.

(See the change in Line 109-131)

6. I'm also missing some explanation on how the average user, not necessarily versed with coding, can make use of the framework. This should be discussed.

Response: We appreciate for the comment. We have put the instructions of the developed

software on the proposed Bio-SoS simulation and process analytical technology in the README.md file that is easy to use by the average users without coding background.

7. Authors state that the python code was included in the .zip file for review. It was not. I would appreciate to have it available when the authors reply.

Response: Appreciate for the comment. The codes have been provided in the revision file.

Reviewer #2

1. This article discussed the development of a framework for considering 3-dimensional multiscale models of induced pluripotent stem cell (iPSC) aggregate dynamics. The authors highlight the complexity of these multiscale processes due to large spatial heterogeneities across a large aggregate system. They introduce a modular framework that incorporates stochastic models of reaction networks, a reaction-diffusion model for the bioreactor conditions, and a population balance model for the cell population. The framework is composed of Stochastic Partial Differential Equations (SPDEs) that describe the evolution of the mass/volume density of cell aggregates over space and time. This work only considers the coalescence of aggregates merging at a size dependent rate (not splitting or non-spherical shape changes) based on an empirical model function. The stochasticity is implemented as Poissonian distributed reaction events within annuli areas of the aggregate. However, it is unclear what the main contribution of this article is. It appears that the reaction modelling of a companion paper (reference 21 in the article) has been coupled to a spatially dependent reaction term of a reaction-diffusion PDE where the spatial dependency is determined by a second PDE representing aggregate growth. The simulations results were compared to experimental results (also from reference 21). The main contribution could be made more explicit.

Response: We appreciate for the comment. In the revision, we improve the presentation and clearly describe the key contributions of this study. In specific, the novelty is summarized below.

- a) Our approach is a first attempt of developing a multiscale bioprocess mechanistic model, called the biological system-of-systems (Bio-SoS), for iPSC monolayer and aggregate cultures, characterizing the causal interdependencies at molecular, cellular, and macro-scope levels. It integrates intracellular, intra-aggregate, and medium bulk culture conditions. Given any feeding protocol and media compositions, it allows us to predict the iPSC culture process outputs (i.e., yield and cell production quality variation) across a wide range of conditions.

- b) The proposed Bio-SoS model has modular design. It includes three modules: (1) single-cell stochastic metabolic network model describing cell response to environmental variation; (2) population balance model characterizing cell-to-cell interaction and aggregation process; and (3) reaction-diffusion model characterizing spatial heterogeneity of micro-environments. This modular design facilitates assembling different production processes, which can accelerate the production process development for iPSC large-scale manufacturing without extensive experiments. Fig 4 illustrates that the Bio-SoS model can use the data of 2D monolayer cultures in petri-dish (that are often collected in the Lab) to provide reliable predictions of 3D aggregate cultures in suspension bioreactors recommended for real large-scale industrial production. Instead of performing an extensive number of expensive and complex aggregate culture experiments in bioreactors and measuring the metabolite concentration within aggregates, the Bio-SoS model can be estimated using data from simpler, more cost-effective monolayer cultures in petri-dish.

Therefore, the system of modules can be a) assembled to facilitate data integration and improve the prediction of monolayer and aggregate cultures; and b) utilized to control cell product quality heterogeneity and optimize the production process performance. Also, the modular design of the Bio-SoS model will facilitate future extensions to incorporate cell responses (i.e., metabolic flux rates, phenotype, gene expression, pluripotency) to both mechanical (e.g., stirred speed, hydrodynamic force, shear stress) and chemical (e.g., concentrations of nutrients/metabolites/oxygen/ions, pH) environmental conditions.

- c) To enhance computational efficiency, a problem often encountered with other simulation software/tools, the simulation complexity of the Bio-SoS model does not increase as the manufacturing scale or bioreactor size increases due to various reasons, including: (1) a coarse-grained approach that divides each aggregate into small spherical shells and assumes that micro-environments and cellular metabolisms are homogeneous in each spherical shell; and (2) a single-cell stochastic metabolic network module characterizing cell response to environmental variation with the distribution of micro-environment condition shaped by cell-to-cell interaction. That means we do not need to separately model and simulate each cell, instead the distribution of cells experiencing different micro-environmental conditions. Therefore, the Bio-SoS can improve the computational efficiency and support the process control for large-scale iPSC manufacturing.

In specific, cell aggregates were divided to L groups with different size. Each aggregate is divided to N_ℓ spherical shells. Consequently, there are $\left(\frac{1}{L}\sum_{\ell=1}^L N_\ell\right)$ number of partitions with each corresponding to specific homogeneous micro-environmental condition. We have a single-cell stochastic metabolic network model that can characterize cell response to different environmental conditions. This fixed number of partitions doesn't change with the bioreactor's size, ensuring that Bio-SoS scales effectively for large-scale industrial applications. Consequently, our framework achieves a significant improvement in computational efficiency compared to the general multiscale simulation. The overall

simulation time for a 72-hour iPSC culture of millions of cells is about 0.1 CPU hour.

- d) In the biopharmaceutical manufacturing, it is critical to ensure the product quality consistency. Since the spatial heterogeneity can contribute to the cell product quality variation, we derive a variance decomposition analysis on the Bio-SoS mechanistic model to quantify the contribution of aggregates with different size to the cell metabolic heterogeneity. The proposed variance decomposition analysis improves the understanding of spatial heterogeneity within iPSC aggregate culture and guides the control of aggregate size, which can avoid unwanted cell death and ensure product quality consistency during cell culture and expansion process.

In addition to methodological novelty, this paper also presents the key contributions for iPSC culture and cell processing. That means the proposed Bio-SoS framework: (1) has the potential to advance the understanding of underlying mechanisms; (2) predicts the impact of critical factors (i.e., aggregate size) on iPSC metabolic heterogeneity; and (3) provides a valuable tool for yield optimization and product quality consistency control.

(See the change in Line 109-131)

2. The authors motivate the need for this framework due to the need to develop modelling paradigms to inform clinical work. In particular, the authors claim that "... a comprehensive model that captures the multi-scale and heterogeneity nature of iPSC aggregate cultures is still lacking". Multiscale models for similar aggregate structures (mammalian aggregates, cancer tissue studies, biofilms) have been made, (see PhysiCell, Chaste, ChemChaste, BMX, Morpheus). Under the umbrella term of agent-based models, these software packages couple cell-based metabolic reaction networks with reaction-diffusion and individual cell growth dynamics. Additionally, these models can incorporate specialized stochastic cell models to represent regulatory structures and cell-cell physical dynamics through calculating the forces acting on a cell. (1) While strong arguments can be made to forego such computational complexity in models for iPSC aggregates, to not mention these agent-based models seems like an oversight. Specifically, in this case where the authors propose a multi-scale 'mechanistic' model considering processes at the individual cell, aggregate, and cell population levels; all aspects handled by the agent-based software packages. (2) This potential gap in the narrative is only amplified when the authors write "... the Bio-SoS framework marks a revolutionary step in the mechanistic model development of multi-scale bioprocess...", albeit written in the discussion. This statement is somewhat hyperbolic as there is little evidence to suggest that the proposed framework "effectively characterizing cell-to-cell interactions and complex mechanisms of iPSC aggregate culture" as no physical interactions are considered.

Response: Thanks for Reviewer's valuable comments. In the revised paper, we added the reviews and discussion on the studies (i.e., PhysiCell, Chaste, ChemChaste, BMX, Morpheus). In addition, we also rewrite the sentence "... the Bio-SoS framework marks a revolutionary step in the mechanistic model development of multi-scale bioprocess..." mentioned by the reviewer to

“The introduction of the Bio-SoS framework marks an important step in the development of multi-scale bioprocess mechanistic model and analytical technology for iPSC cultures.”

Differing with the clinical studies (such as tissue studies and biofilms) and associated modeling and software tools mentioned by the reviewer, we want to clarify this study is motivated by the critical challenges and needs in the biopharmaceutical manufacturing process development and control, specifically iPSC cultures for large-scale manufacturing cell therapies and regenerative medicines, including: (1) integrate the data from different production processes and accelerate the process development to support iPSC manufacturing scale-up without conducting extensive and expensive experiments especially in large-size bioreactors; and (2) improve productivity and ensure cell product quality consistency.

We have taken the reviewer’s feedback into account and have made significant improvements to our paper presentation. In the revision, we have reviewed existing multiscale models and simulation software mentioned by the reviewer.

- PhysiCell⁹ is an open-source physics-based multicellular simulator. It is an agent-based modeling framework accounting for mechanically and biochemically interacting cells in dynamic tissue microenvironments with multiple diffusing substrates and cell secreted signals. It focuses on cell-level simulation and does not account for intracellular metabolism so far.
- BMX¹⁰ is a software system developed for the high-performance modelling of large bacteria cell communities by utilizing GPU acceleration. It builds on AMReX adaptive mesh refinement package, aimed at efficiently modeling cell colony formation under realistic laboratory conditions. This software needs to expand the capabilities for experimental studies and include a sufficient model for the bacteria cell metabolism.
- Morpheus¹¹ is a user-friendly modeling environment with strong capacity in multiscale and multicellular systems simulation. However, for very large and complex system such as iPSC bioreactor cultures, computational speed and memory could become an issue.
- Chaste¹² is a cell-based modelling framework for simulating biological tissues and understanding the process of tissue growth and repair. This simulation software is based on continuous mechanics PDE/ODE equations and finite element method, but it does not account for concurrent bulk and intracellular biochemical reactions.
- ChemChaste¹³ is computational framework for hybrid continuum-discrete modeling of multicellular populations (i.e., microbial communities) coupled through chemical reaction-diffusion systems. It is an advanced simulation tool where a cellular mesh is defined wherein each mesh node acts as the center of a cell and each cell is simulated as a particle.

It's important to note that while existing modeling approaches and software packages described above have proven to be effective for simulating small aggregate systems, they are often too computational expensive when it comes to simulating the bioreactor iPSC aggregate cultures at an industrial manufacturing scale, especially for manufacturing scaleup. To address this limitation, we have focused on enhancing our model's scalable capacity to accurately simulate the complex iPSC aggregate cultures in bioreactor systems, advancing the understanding on more practical iPSC cultures and manufacturing processes.

In contrast to these existing software packages, Bio-SoS stands out as a specialized coarse-grained simulation tool tailored specifically for both iPSC monolayer and aggregate cultures. Its primary strength lies in its ability to: (1) integrate the data from different production processes; (2) assemble and accelerate the development of iPSC production processes without extensive experiments; and (3) greatly enhance simulation efficiency through a nuanced exploration and application of biological mechanisms. This improvement is achieved by key features:

Explicit Mechanism Modeling for iPSC Aggregation: The Bio-SoS employs a novel approach by explicitly modeling the aggregation mechanism of iPSCs using population balance equations. This allows for a more precise representation of the cellular dynamics during the aggregation process, resulting in a more accurate simulation.

Modular Design for Data Integration: The Bio-SoS incorporates a modular design that simplifies the need for conducting extensive number of expensive and complex aggregate culture experiments in bioreactors, which enables model parameter estimation through data obtained from simpler and more cost-effective 2D monolayer cultures in petri-dish instead of bioreactor cultures, as well as from other sources, such as literature studies on different iPSC production processes.

Homogeneous Cellular Metabolism Assumption: To reduce the simulation cost within aggregates, the Bio-SoS simplifies the representation of cellular metabolisms by assuming homogeneous cellular metabolisms within each spherical shell / annulus of the aggregate. This assumption helps to reduce computational complexity while maintaining a sufficient level of accuracy in the simulation.

(See the change in Line 84-92)

3. In contrast the authors model aggregates as a series of annuli that assumes each cell across the aggregate annulus has the same phenotype and behavior despite the discussed heterogeneous conditions. The authors demonstrate that the stochastic model may be reduced to a PDE model with metabolic flux analysis representing the reaction terms. The authors also only average over 10 simulation outputs. Therefore, the evidence to support the need to consider stochastic models is weak and the error due spatial assumptions may override a stochastic signal. For the concentrations of the metabolites found in a cell, is a stochastic mathematical model necessary? Are the authors predicting a phenomenon that could not be achieved through deterministic models of chemical reactions? These aspects would need to be discussed to provide support to the framework. This framework may find some success as a coarse-grained simulation of aggregate structures. The simulation time will be quicker than more complex models and may be more suitable for an industrial application.

Response: We appreciate Reviewer's comments. In the revision, we improve the presentation to address the associated confusion and concerns. Here, we provide the point-to-point response below.

First, it is essential to clarify that the primary focus of this paper lies in investigating the spatial and metabolic heterogeneity of iPSC aggregates. Due to strong cell-to-cell interactions, iPSCs can form large cell aggregates in suspension bioreactors, resulting in spatial heterogeneous micro-environment conditions, i.e., insufficient nutrient supply and extra metabolic waste build-up for the cells located at core. To understand and predict cell metabolic response to different environmental conditions that can occur in the aggregates, a PDE based metabolic kinetic model was developed in Wang et al. (2023)². Suppose cells cultured in petri-dish 2D monolayer have homogeneous environment condition. Thus, this deterministic PDE mechanistic model characterizes the mean metabolic flux rate response for homogeneous cell population. It was fitted and validated by utilizing data gathered from well-designed monolayer K3 iPSC cultures in Odenwelder et al. (2021)³. Built on this backbone mean reaction mechanistic model, by following the recent studies on stochastic reaction network, e.g., Kloska et al. (2019, 2022)^{7,8}, we construct a stochastic metabolic network (SMN) for single cells that can characterize: (1) the stochastic reaction network for individual cells under the same environmental condition; and (2) cell metabolic response to environmental change. Therefore, for 2D monolayer culture with given initial environmental condition, the results in Fig 3 demonstrate that the expected metabolic dynamics for homogenous cell population predicted by the Bio-SoS match well with the PDE deterministic mechanistic model.

Second, to address this concern “The authors also only average over 10 simulation outputs”, in the revision, we have increased the number of simulations runs to 30 and have found that the results remain highly robust.

Third, while modeling metabolic reaction stochastic uncertainty or noise for homogeneous cells under the same environmental condition is not the central objective of this study, retaining stochastic elements in our simulation serves to enhance the versatility of our model. It positions us for future research into predicting iPSC aging, especially as we consider integrating gene expression into our multi-scale bioprocess Bio-SoS model. As suggested by Raj and Alexander (2008)⁵, noise in gene expression is generally undesirable and has been shown to correlate with aging. For instance, one research study⁶ demonstrated that the expression levels of various housekeeping and cell-type-specific genes in individual murine cardiac myocytes became increasingly stochastic as the organism aged.

Fourth, through integrating a coarse-grained simulation of aggregate structures, the Bio-SoS model uses one single-cell stochastic metabolic network (SMN) module to quantify the distribution of cells in different micro-environmental conditions. The model complexity does not increase with the number of cells, which can support the prediction and real-time control for large-scale manufacturing process.

Specific comments:

1. The framework code was not provided.

Response: We appreciate for the comment. The codes have been provided in the revision file.

2. Further proof reading by the authors is recommended (see annotated .pdf).

Response: We appreciate for the comment. We have corrected the issue.

3. Line 194, “which provided an excellent 194 opportunity to predict iPSC culture behavior not used in the training of the model.” It’s not clear what this training entails, were core model parameters optimized for and applied to future model experiments?

Response: The training refers to the model parameter fitting. The Bio-SoS model was fitted by utilizing 2D monolayer culture data from K3 iPSCs and cell aggregation profiles of hESCs from Wu et al. (2014)¹. Then the fitted Bio-SoS model was used to predict the iPSC bioreactor culture behavior observed in Kwok et al. (2018)⁴ (which is not used for fitting the model parameters).

4. The excretion of extracellular matrix materials was mentioned in the introduction, but the cell/aggregate models do not exhibit such a capability.

Response: We appreciate the reviewer’s comment. In the revised paper, we removed the description of excretion of extracellular matrix materials.

5. Similarly, how was the ‘validation’ for the model from the data performed? Statistical or qualitative validation?

Response: The details of the model validation can be found in Results section. Here we summarize the steps of validation:

- a) Population balance model (PBM) was validated using cell proliferation and aggregation dynamics data from Wu et al. (2014)¹. The experimental observations and model predicted aggregate growth profiles are shown in Fig. 2a and the time-varying aggregate size distribution is illustrated in Fig. 2b. The observed and predicted values exhibit a strong consistency, thereby indicating a good prediction performance of the model.
- b) The porosity and tortuosity of reaction-diffusion model (RDM) were validated by using stirred-tank suspension bioreactor data from Wu et al. (2014)¹. The diffusion coefficients of extracellular metabolites are adapted from various literatures (see details in Supplementary Table 4).
- c) The metabolic dynamic mean response model parameters of stochastic metabolic network (SMN) for homogeneous cell population came from the iPSC culture kinetic model described in Wang et al. (2023)². To calibrate the model parameters, we utilized 2D monolayer iPSC culture data provided by Odenwelder et al. (2021)³ (an experimental study conducted by the research team led by a corresponding author of this paper). These cells cultured in 2D monolayer within a petri-dish can be considered to be homogeneous for several reasons: (1) cells in a 2D monolayer are in physical contact with their neighboring cells on a flat surface, promoting

uniform cell-to-cell interactions and (2) all cells are exposed to the culture medium's nutrients, gases, and soluble factors uniformly.

- d) To further validate the performance of the Bio-SoS model for homogeneous cell population from 2D monolayer cultures, we carried out a comparative analysis between the population metabolic dynamics predicted by the Bio-SoS model and those generated by the mechanistic model developed in Wang et al. (2023)², focusing specifically on 2D monolayer cultures of K3 iPSCs. The results, which are displayed in Fig. 3, demonstrate strong alignment between the predicted values from the Bio-SoS model and the mechanistic model from Wang et al. (2023).
 - e) To test the extrapolation prediction performance of the Bio-SoS model, we used stirred suspension culture data of FSiPS (short for FS hiPSC clone 2) from Kwok et al. (2018)⁴. Following the same culture protocol (i.e., feeding strategy), the predicted values of the Bio-SoS for glucose consumption and lactate production were visually compared to the measured values. The predictions closely align with the experimental data, falling within the 95% confidence intervals after rescaling the vertical axes; see Fig. 4.
6. The metabolic model only considers enzymatic reactions in a kinetic manner, using Monod kinetics. The biomass growth flux objective is the product of the amino acid synthesis and does not consider other objective products such as cell wall or protein synthesis as used in more complex biomass objectives. This metabolic model does not account for enzyme cost or higher control structures of the cell. A comment on the production of the biomass objective would be appreciated.

Response: Thanks for the comment. In the Bio-SoS model, we do not use the biomass growth flux as objective. In addition, the iPSCs only have cell membranes, and cell membranes are mainly composed of proteins and lipids. The **proteins** are represented in the model by **amino acid fluxes** and the **lipids** are represented in the model by **"0.16Glc"** in the biomass stoichiometric equation as follows used in the paper,

For further discussion, please refer to Odenwelder et al. (2021)³. Compared to bacteria and yeast, iPSCs have very limited range of culture condition where the cells are still functional and useful at the end of the culture. Thus, there is no reason to model iPSCs under low-energy or other extreme conditions.

7. Figure 6 is difficult to read.

Response: Thanks for the comment. We have improved Fig. 6. by reordering the reactions.

8. Pages 13 and 14, which figure is denoted as 'Figure 11' is unclear. The figures and captions look muddled, line 312 looks like a caption?

Response: Thanks for the comment. There are cross-reference issues in the .docx file, which cause the pdf conversion fails. We have fixed all related issues.

9. A finite difference of grid space 15 micrometer and aggregate radius in steps of 60 micrometer, leaving only 4 finite grid difference. Is this sufficient? The diameter of a cell is also 15 micrometer. The optimal aggregate was seen to be 10 grid cells in radius, is this step size precise enough for accurate simulation? What is the temporal step size to ensure numerical stability (Courant-Friedrichs-Lewy condition) of PDEs?

Response: Thanks for commenting the step size. We apologize for the typo. We corrected the description about grid space in our revision. In our implementation, the population balance model was solved numerically by finite differences over equally spaced grid of 1 μm and 0.1 hour for spatial-temporal grid size.

10. Lines 174-177. The analysis presented ties the aggregate dynamics as a balance between resource consumption in the interior and buildup of metabolic waste products. In both experiments (Liu, J. et al. (2015).) and models (Bocci, F. et al. (2018).) on the similar biofilm system, this same interplay was seen to induce film/aggregate radius oscillations, was this seen in the model or can the authors comment on whether oscillating dynamics were/could be seen in the aggregate model? Shrinking of the aggregate would require cell death which, while mentioned, does not seem to be included in the growth model.

Response: We appreciate the reviewer's feedback. It should be noted that biofilms are typically formed by bacteria, not iPSCs. Differing with biofilms in the clinical study, we have never observed radius oscillation in iPSC aggregate and our cell counting method records the population size distribution.

Based on the iPSC culture data utilized in this study, the shrinking of iPSC aggregates has not been observed. Consequently, we do not have the experimental observation on either oscillations or the shrinking of iPSC aggregate size. For example, a study by Kwok et al. (2018)⁴ recorded the diameters of iPSC aggregates in bioreactor cultures over a 7-day period and found a statistically consistent increase in aggregate sizes.

In addition, the iPSC model was constrained to culture conditions that enable maintaining functional and useful cells at the end of the culture. Based on our limited knowledge on biofilms in the clinical study, the purpose of modeling biofilms is to either enable cleaning surgical devices or to understand disease resistance in patients, such that the biofilm can be removed or destroyed. Therefore, iPSC and biofilm models consider very different situations and have different objectives.

Despite the limitation in the iPSC aggregate culture data, if the experimental observation indicates the cell death impact, our model can be extended to account for the shrinking of the iPSC aggregates due to the cell death, i.e.,

$$\frac{\partial x}{\partial t} = \alpha_G \cdot x \log\left(\frac{M}{x}\right) - d(x)$$

where $d(x)$ represents the impact of aggregate size decrease associated to some cell death equation. In the revised paper, we added the discussion on it.

Reference

1. Wu J, Rostami MR, Cadavid Olaya DP, Tzanakakis ES. Oxygen transport and stem cell aggregation in stirred-suspension bioreactor cultures. *PLoS One* **9**, e102486 (2014).
2. Wang K, Xie W, Harcum SW. Metabolic Regulatory Network Kinetic Modeling with Multiple Isotopic Tracers for iPSCs. arXiv preprint arXiv:230500165, (2023).
3. Odenwelder DC, Lu X, Harcum SW. Induced pluripotent stem cells can utilize lactate as a metabolic substrate to support proliferation. *Biotechnology Progress* **37**, e3090 (2021).
4. Kwok CK, *et al.* Scalable stirred suspension culture for the generation of billions of human induced pluripotent stem cells using single-use bioreactors. *Journal of tissue engineering and regenerative medicine* **12**, e1076-e1087 (2018).
5. Raj, A., & Van Oudenaarden, A. (2008). Nature, nurture, or chance: stochastic gene expression and its consequences. *Cell*, 135(2), 216-226.
6. Bahar, R., Hartmann, C.H., Rodriguez, K.A., Denny, A.D., Busuttill, R.A., Dollé, M.E., Calder, R.B., Chisholm, G.B., Pollock, B.H., Klein, C.A. and Vijg, J., 2006. Increased cell-to-cell variation in gene expression in ageing mouse heart. *Nature*, 441(7096), pp.1011-1014.
7. Sylwester M. Kloska, Krzysztof Pałczyński, Tomasz Marciniak, Tomasz Talaśka, Marissa Miller, Beata J. Wysocki, Paul Davis & Tadeusz A. Wysocki, Queueing theory model of pentose phosphate pathway, *Scientific Reports*, Scientific Reports volume 12, Article number: 4601 (2022).
8. Sylwester Kloska, Krzysztof Pałczyński, Tomasz Marciniak, Tomasz Talaska, Marissa Nitz, Beata J. Wysocki, Paul Davis, and Tadeusz A. Wysocki. Queueing theory model of Krebs cycle. *Bioinformatics*, 37(18), 2021, 2912–2919.
9. Ghaffarizadeh A, Heiland R, Friedman SH, Mumenthaler SM, Macklin P. PhysiCell: An open source physics-based cell simulator for 3-D multicellular systems. *PLoS Comput Biol* **14**, e1005991 (2018).
10. Palmer BJ, Almgren AS, Johnson CGM, Myers AT, Cannon WR. BMX: Biological modelling and interface exchange. *Scientific Reports* **13**, 12235 (2023).
11. Starruss J, de Back W, Bruschi L, Deutsch A. Morpheus: a user-friendly modeling environment for multiscale and multicellular systems biology. *Bioinformatics* **30**, 1331-1332 (2014).
12. Cooper FR, et al. Chaste: Cancer, Heart and Soft Tissue Environment. *J Open Source Softw* **5**, 1848 (2020).
13. Johnson CGM, Fletcher AG, Soyer OS. ChemChaste: Simulating spatially inhomogeneous biochemical reaction-diffusion systems for modeling cell-environment feedbacks. *Gigascience* **11**, (2022).

REVIEWERS' COMMENTS:

Reviewer #1 (Remarks to the Author):

The authors have addressed all my comments.

Reviewer #2 (Remarks to the Author):

The authors have provided valuable insight and have demonstrably actioned all review comments in their response, editing the manuscript appropriately. The updated manuscript is clear and demonstrates a robust and detailed modelling suite that can provide valuable insights for the pharmaceutical engineering field. I have no more queries or reservations about the quality or applicability of this work.

Reviewer #1

The authors have addressed all my comments.

Response: Appreciate for the comment.

Reviewer #2

The authors have provided valuable insight and have demonstrably actioned all review comments in their response, editing the manuscript appropriately. The updated manuscript is clear and demonstrates a robust and detailed modelling suite that can provide valuable insights for the pharmaceutical engineering field. I have no more queries or reservations about the quality or applicability of this work.

Response: Appreciate for the comment.